# Low Rank Gradients and Where to Find Them

**Rishi Sonthalia**[*]
Department of Mathematics
Boston College
rishi.sonthalia@bc.edu

**Michael Murray**
Department of Mathematics
Bath University
mjm253@bath.ac.uk

**Guido Montúfar**
Mathematics and Statistics & Data Science
University of California, Los Angeles
montufar@math.ucla.edu

## Abstract

This paper investigates low-rank structure in the gradients of the training loss for two-layer neural networks while relaxing the usual isotropy assumptions on the training data and parameters. We consider a spiked data model in which the bulk can be anisotropic and ill-conditioned, we do not require independent data and weight matrices and we also analyze both the mean-field and neural-tangent-kernel scalings. We show that the gradient with respect to the input weights is approximately low rank and is dominated by two rank-one terms: one aligned with the bulk data-residue, and another aligned with the rank one spike in the input data. We characterize how properties of the training data, the scaling regime and the activation function govern the balance between these two components. Additionally, we also demonstrate that standard regularizers, such as weight decay, input noise and Jacobian penalties, also selectively modulate these components. Experiments on synthetic and real data corroborate our theoretical predictions.

## 1 Introduction

Feature learning is a critical driver behind the success of deep learning. Despite this, a theoretical characterization of it remains elusive. In order to drive understanding, a line of research [3, 12–14, 28, 45] has emerged studying two-layer networks whose inner weights are trained or updated via one step of gradient descent. In this context, feature learning can be characterized through the emergence of a low-rank structure in the network weights. In particular, when the weights of many of the neurons align in a predominant direction, the matrix of weights becomes approximately low rank. Moreover, Ba et al. [3] proved that a ridge estimator trained on such features can outperform random feature models and other kernel methods. However, these prior investigations require idealized conditions, for example isotropic data or weights, which diverge from real-world scenarios where data typically exhibits anisotropy or an ill-conditioned covariance. In addition, the effects of regularization in this context have also been underexplored.

This paper addresses two questions: *1) how do low-rank gradient phenomena arise and behave under more general conditions of anisotropy and ill-conditioning?* and *2) what impact do common regularizers have on feature learning in this context?* Our analysis accommodates spiked data with an anisotropic ill-conditioned bulk. This allows us to explore the effect of the size of the data spike, controlled by a parameter $\nu \geq 0$, as well as spectral decay profiles of the bulk, controlled by a parameter $\alpha \geq 0$. Our central finding is that the gradient of the inner-layer weights is generically well approximated by a **rank-two matrix**. This structure arises from the interplay of two primary rank-one components: $S_1$, driven by the input bulk and target residue, and $S_2$, driven by the leading

---

[*]Corresponding Author

39th Conference on Neural Information Processing Systems (NeurIPS 2025).

eigenvector of the data covariance. The relative prominence of these components, and consequently the direction of feature learning, is determined by the interplay of data properties, the scale of the network parametrization, the choice of loss and activation function as well as the use of regularization. We corroborate our theoretical findings with experiments on both synthetic data (Sections 3, 3.2 and 4) and real data (MNIST, CIFAR-10 embeddings).[2] A summary of our key contributions is:

- **Generalized Theory of Low-Rank Gradients:** We provide a theoretical framework (Section 3, Theorems 3.1 and 3.2) characterizing the low-rank structure of the gradient under significantly relaxed assumptions on data and weight matrices (anisotropy, ill-conditioning; Section 2).
- **Identification of a Dominant Rank-Two Structure:** We show (Theorems 3.1 and 3.2) that the gradient is often better approximated by a rank-two matrix than the rank-one structures identified in prior specialized settings. We provide conditions under which each of these components dominates.
- **Modulation by Activation Function and Regularization:** We show how activation functions and common regularizers selectively modulate the components of the gradient. We reveal that ReLU can suppress the contribution from the residue $S_1$ (Section 3.2), while input noise and a Jacobian penalty can promote the residue component and data spike component (Section 4) respectively.
- **Mean Field (MF) versus Neural Tangent Kernel (NTK) scaling:** We demonstrate differences in dominant spike alignments, $S_1 \sim X_B^T y$ in MF vs. $S_1 \sim X_B^T r$ in NTK, at initialization (Section 3) and the subsequent impact during training.

## 1.1 Related work

**Low rank gradients in two layer networks:** For a Mean Field (MF) like regime, prior work has shown that the gradient is approximately rank one [3, 13], which results in an alignment between the leading eigenvector of the hidden feature kernel with the target [45]. Dandi et al. [14] showed that to learn $k$ directions, as opposed to a single direction, we need high sample complexity ($n = \Omega(d^k)$). Under an NTK scaling Moniri et al. [28] showed that a learning rate which grows with the sample size introduces multiple rank-one components in the hidden feature kernel. However, these results rely on well-conditioned input data and weight matrices. To ameliorate this issue a number of works have also incorporated ill-conditioning via a spiked covariance models ($\mathcal{N}(0, I + n^\nu qq^T)$) with single-index targets ($\sigma_*(\langle \beta_*, x \rangle)$) [4, 29]. , Ba et al. [4] found dominant rank-one gradients aligned with the data spike $q$ (if aligned with the target $\beta_*$ and $\nu > 1/2$), enabling efficient learning, whereas Mousavi-Hosseini et al. [29] showed that gradient flow might yield weights nearly orthogonal to $q$, even under seemingly favorable conditions ($\nu = 1, \beta_* = q$). *Our work continues this line of research, providing results for more general anisotropic and ill-conditioned data and weight matrices as well the effects of regularization.*

Understanding the spectral evolution of the network's weight and features matrices, particularly the 'bulk' components beyond dominant spikes, remains challenging. While significant progress has been made in characterizing spectra at initialization [1, 6, 16, 19, 30, 32, 34, 44] and after a single step of gradient descent [12], the dynamics over longer timescales are complex.

**Convergence to low-rank weights:** While our analysis focuses on the *gradient updates* that drive learning, related studies investigate the implicit bias of gradient-based optimization towards low-rank solutions [17, 21, 27, 33, 41]. Our findings complement this body of work by characterizing the generically dominant *rank-two* structure within the gradient updates themselves, providing insight into the mechanisms potentially driving this convergence.

## 2 Setup and Assumptions

In this section, we provide the technical details required for analysis. A summary of notation and discussion of examples of when the assumptions hold can be found in Table 1 and Appendix B. We consider shallow networks with $d$ input dimensions, $m$ hidden neurons, and $n$ training data points.

**Assumption 1** (Proportional scaling). *Let $\psi_1, \psi_2 \in \mathbb{R}_{>0}$ be fixed constants. We consider $m, n$ as functions of $d$ such that $n/d \to \psi_1 < 1$ and $m/d \to \psi_2$ as $d \to \infty$.*

**Data:** We consider random input data $x_i \in \mathbb{R}^d$ for $i \in [n]$, sampled i.i.d. These are stored row-wise in a matrix $X \in \mathbb{R}^{n \times d}$. For each $x_i$, the corresponding label is $y_i \in \mathbb{R}$, and labels stored as $y \in \mathbb{R}^n$.

---

[2]All code is available at the anonymous Github repository: https://github.com/rsonthal/Low-Rank-Gradient

**Assumption 2** (Input features distribution). *Let $\hat{\Sigma} \in \mathbb{R}^{d \times d}$ for which there exists an $\alpha \geq 0$ such that the $k$-th eigenvalue satisfies $\lambda_k(\hat{\Sigma}) = k^{-\alpha}$ for $k = 1, \ldots, d$. Let $q \in \mathbb{S}^{d-1}$ and define $\zeta = n^{\nu}$ for some $\nu \geq 0$. We assume each input data point $x_i$ is sampled i.i.d. from a multivariate Gaussian distribution $N(0, \Sigma)$, where the full covariance $\Sigma \in \mathbb{R}^{d \times d}$ is given by $\Sigma = \hat{\Sigma} + \zeta^2 q q^T$.*

Assumption 2 models a bulk component via $\hat{\Sigma}$ and a spike component via $q$ (magnitude $\zeta$) and allows general forms of ill-conditioning with $\lambda_d(\hat{\Sigma}) \to 0$ if $\alpha > 0$, and $\lambda_1(\Sigma) \to \infty$ if $\nu > 0$. This generalizes typical data distribution assumptions like isotropic Gaussian ($\Sigma = I_d$) or uniform on a sphere [3, 14, 28, 31, 45], anisotropic data with a bounded condition number [15, 18], divergent largest eigenvalue and bounded smallest eigenvalue [4, 22, 24, 29, 40], or bounded largest eigenvalue and decaying smallest eigenvalue [5, 8, 43].

**Network:** We consider a two-layer neural network with input-output map $f : \mathbb{R}^d \to \mathbb{R}$ defined as

$$f(x) = \gamma_m a^T \sigma(Wx) \in \mathbb{R}. \tag{1}$$

Here, $W = [w_1, \ldots, w_m]^T \in \mathbb{R}^{m \times d}$ is the matrix of inner (first-layer) weights, $w_j \in \mathbb{R}^d$ is the weight vector for the $j$-th hidden neuron and $a \in \mathbb{R}^m$ is the vector of outer (second-layer) weights. The activation function $\sigma : \mathbb{R} \to \mathbb{R}$ is applied element-wise to the preactivations $Wx$. The parameter $\gamma_m \in \mathbb{R}_{>0}$ is a non-trainable scaling constant that depends on the network width $m$.

**Assumption 3** (Network parameters). *We assume the following for $W, a$ and $\gamma_m$:*

1. ***Outer weights:*** *Elements $a_j$ are sampled i.i.d. from $\mathrm{Uniform}(\{-1, 1\})$.*
2. ***Inner weights:*** *Rows $w_j$ of $W$ have unit length, $w_j \in \mathbb{S}^{d-1}$.*
3. ***Scaling parameter:*** *$\gamma_m = \Theta(1/\sqrt{m})$ (NTK scaling) or $\gamma_m = \Theta(1/m)$ (MF scaling).*

The assumption on $a$ is standard. The assumption on $W$ (unit-norm rows) relaxes typical literature requirements (e.g., isotropic Gaussian or uniformly spherical $w_j$). This allows modeling anisotropic weights, possibly dependent on $X$, to analyze updates throughout training, not just at initialization. The scaling parameter $\gamma_m$ defines two common regimes: NTK ($\gamma_m \sim 1/\sqrt{m}$) [2, 20, 23, 25], associated with lazy training where inner weights vary little [10, 25], and MF ($\gamma_m \sim 1/m$), associated with feature learning [9, 26, 36, 39]. These scalings yield different initial output variances $(\mathrm{Var}(f(x)) = \Theta(1)$ in NTK vs. $o(1)$ in MF), impacting dynamics.

**Assumption 4** (Activation function). *The activation function $\sigma : \mathbb{R} \to \mathbb{R}$ satisfies:*

1. ***Smoothness:*** *$\sigma'$ and $\sigma''$, first and second derivatives of $\sigma$, exist almost everywhere on $\mathbb{R}$.*
2. ***Lipschitzness:*** *$\sigma$ and $\sigma'$ are $L$-Lipschitz for some constant $L > 0$.*
3. ***Non-trivial expected derivative:*** *For $x \sim N(0, \Sigma)$ (per Assumption 2) and $W$ (per Assumption 3), let $\mu_j = \mathbb{E}_x[\sigma'(w_j^T x)]$ (expectation over $x$ for a given $w_j$). We assume $\mu_j = \Omega(1)$ for all $j$. Let $\mu = [\mu_1, \ldots, \mu_m]^T$. We define $\sigma'_{\perp}(Wx)_j = \sigma'(w_j^T x) - \mu_j$.*

Common activation functions, such as Sigmoid, Tanh, ELU [11], Swish [35], Softplus, satisfy the *Smoothness* and *Lipschitzness*. We note that the derivative of ReLU is not Lipschitz. The condition $\mu_j = \Omega(1)$ (non-vanishing expected derivative) is satisfied by ELU, Swish, and Softplus generically, and for Sigmoid and Tanh as long as $w_j^T \Sigma w_j = O(1)$. See Section B.1 for more details.

**Parameter update via GD:** Let $\ell : \mathbb{R} \times \mathbb{R} \to \mathbb{R}_{\geq 0}$ be a function which measures the loss between a label and a prediction. With $f$ defined as per Equation 1, we define the loss given a dataset $(X, y) = (x_i, y_i)_{i \in [n]}$ with respect to the inner-layer weights $W$ as $L(W) = \frac{1}{n} \sum_{i=1}^{n} \ell(f(x_i), y_i) + \lambda R(W)$. $R$ denotes a regularization function (e.g., the 2-norm $R(W) = \|W\|_F^2$), and $\lambda \in \mathbb{R}_{\geq 0}$ is the regularization parameter. We consider an update to $W$ arising from one step of GD, $W \leftarrow W - \eta \nabla_W L(f(X), y)$, where $\eta > 0$ denotes the step size. We define the *residue* vector as

$$r = [\partial \ell(f(x_1), y_1)/\partial f(x_1), \ldots, \partial \ell(f(x_n), y_n)/\partial f(x_n)]^T \in \mathbb{R}^n. \tag{2}$$

To motivate this terminology, consider that for the Mean Squared Error (MSE) loss, $r$ corresponds to the vector of residues $[f(x_i) - y_i]_i$. More generally, for many losses $r$ can typically be interpreted as the component of the targets not captured by the predictions of the model (see Section B.2).

**Proposition 2.1** (Gradient of the loss). *If Assumption 4 holds and $R$ is differentiable, then*

$$G := \nabla_{W^T} L = \gamma_m X^T \left[ (ra^T) \circ \sigma'(XW^T) \right] + \lambda \nabla_{W^T} R(W) \in \mathbb{R}^{d \times m}$$

*exists for almost every $W$ in $\mathbb{R}^{m \times d}$.*

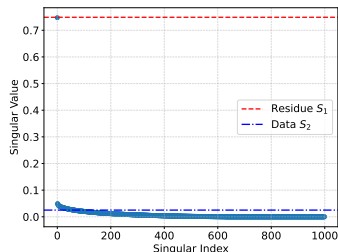 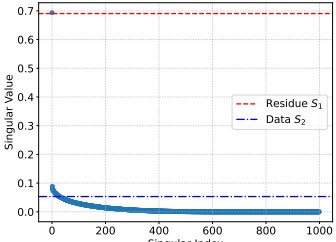 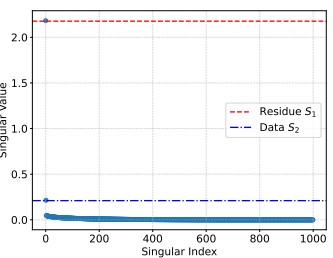

**(a)** Tanh, BCE, $\nu = \frac{1}{8}$, isotropic $W$.  **(b)** Swish, Hinge, $\nu = \frac{3}{8}$, non-isotropic $W$.  **(c)** Softplus, BCE, $\nu = \frac{3}{8}$, non-isotropic $W$.

**Figure 1:** Singular value distribution of the gradient $G$ for varying activation, loss and $\nu$ and weight distribution. Red, and blue lines show the singular value of $S_1$, and $S_2$ respectively. In **(a)** the rows of $W$ are i.i.d. uniformly random on the unit sphere, we denote this $W = W_S$. In **(b)** and **(c)** then $W = W_S + n^{-1/4}\mathbf{1}q^T$, where $W$ is then normalized. The following parameters are constant across all experiments: $\alpha = 0$, $\gamma_m = \frac{1}{\sqrt{m}}$ (NTK) $n = 750, d = 1000, m = 1250$. The targets $y$ are given by a triple index model, see Appendix C. For $\nu < 0.25$, a single residue-aligned spike is seen for both isotropic and non-isotropic $W$. For $\nu \in [0.25, 0.5)$, the gradient is approximately rank two.

For our results to hold we require the following technical assumption on the residues.

**Assumption 5** (Residue concentration). *Under the proportional scaling regime (Assumption 1), with probability $1 - o(1)$ over the training data $(X, y)$, the residue $r$ satisfies*

$$\frac{\|r\|_\infty}{\|r\|_2} = O\left(\frac{\log n}{\sqrt{n}}\right).$$

We emphasize that Assumption 5 is a mild condition: it ensures that no single component of the residue vector disproportionately dominates its overall $\ell_2$ norm. Such a condition typically holds if the residues $r_i$ are i.i.d. subgaussian random variables. See Appendix B.3 for more discussion.

Our analysis also depends on the alignment between the residue $r$ and specific structural components of the input data $X$. From Assumption 2 we have the following decomposition of the input features,

$$X = X_B + X_S = X_B + \zeta z q^T \in \mathbb{R}^{n \times d}, \tag{3}$$

where $X_B$ has rows sampled i.i.d. from $\mathcal{N}(0, \hat{\Sigma})$, $z \sim \mathcal{N}(0, I)$, and $q$ is a unit vector. We note that for sufficiently large $\zeta$, $z$ is approximately the principal eigenvector of $XX^T$. One of the inputs to our analysis will be the degree of alignment between the residue vector $r$ and the spike component $z$ of the input data. The projection of the residue $r$ onto the principal eigenvector of $XX^T$ is a natural statistic of interest and has been considered in prior works [20, 38]. In Appendix B.4 we provide $\beta$ estimates for 192 scenarios.

**Assumption 6** (Residue alignment). *With probability $1 - o(1)$, $\left|\frac{1}{\sqrt{n}\|r\|_2}z^T r\right| = \Theta(d^{-\beta/2})$.*

## 3 Spiked Data Leads to a Low-Rank Gradient

In this section we analyze the role of spiked data in shaping the gradient with no explicit regularization, $\lambda = 0$. We demonstrate that for a spiked data covariance the gradient $G$ is either approximately rank one or rank two, depending primarily on the size of the spike. To demonstrate this we define the following three rank-one matrices:

**Residue Spike:** $S_1 := \frac{\gamma_m}{n}\left(X_B^\top r\right)(a \circ \mu)^\top$,  **Data Spike:** $S_2 := \frac{\gamma_m \zeta}{n} q\left[z^\top((ra^\top) \circ \sigma_\perp'(XW^\top))\right]$,

**Interpolant:** $S_{12} := \gamma_m \zeta \frac{z^\top r}{n} q(a \circ \mu)^\top$.

We remark that $S_1$ is studied in [3] and $S_2$ is analogous to the gradient update in [4]. The matrix $S_{12}$ interpolates between the two: in particular, $S_{12}$ and $S_1$ have the same right singular vector and $S_{12}$ and $S_2$ have the same left singular vector. Hence $S_1 + S_{12}$ and $S_{12} + S_2$ are both rank one.

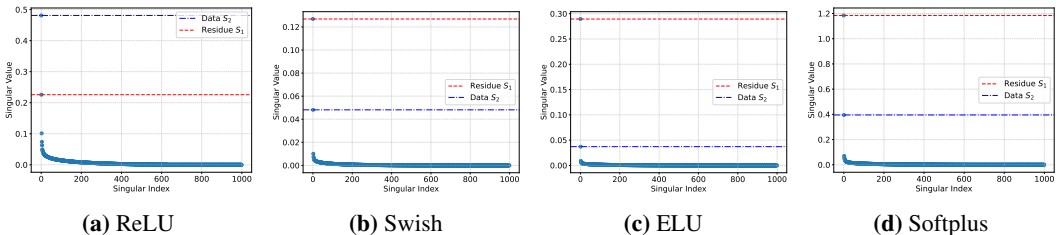

| **(a)** ReLU | **(b)** Swish | **(c)** ELU | **(d)** Softplus |

**Figure 2:** ReLU suppresses the residue spike ($S_1$) compared to smooth activations. Fixed parameters: $\nu = 1/8$, $\alpha = 5/9$, $n = 750$, $d = 1000$, and $m = 1250$.

### 3.1  Small-to-Moderate Spike ($\nu \in [0, 0.5)$): $\mathcal{C}^2$ Activations

The key contribution of this section is Theorem 3.1, which characterizes the approximate low rank structure of the gradient for small-to-moderate spike sizes. We also note that Theorem 3.1 generalizes [3, Proposition 2] by covering a broader range of covariance structures, loss functions and initialization scalings[3]. In the small spike setting, $\nu \in [0, 1/4)$, the gradient is approximately rank one and aligns with the residue plus interpolant $S_1 + S_{12}$. By contrast, in the moderate spike setting, $\nu \in [1/4, 1/2)$, the gradient becomes rank two. We empirically verify these our theoretical results (Figures 1) across a range of activation and loss functions under the NTK scaling.

**Theorem 3.1** (Gradient approximation). *Suppose Assumptions 1, 2, 3, 4, 5, 6 are satisfied, $X$ and $W$ are independent, and $\sigma$ is a $\mathcal{C}^2$ function. Define $E = G - S_1 - S_{12} - S_2$. Then, for all $\nu, \alpha \in \mathbb{R}_{\geq 0}$,*

$$\frac{\|G - S_1 - S_{12}\|_2}{\sqrt{m}\gamma_m\|r\|_\infty} = O\left(\|W\|_2 n^{2\nu - \frac{1}{2}}\right), \qquad \frac{\|G - S_1 - S_{12} - S_2\|_2}{\sqrt{m}\gamma_m\|r\|_\infty} = O\left(\|W\|_2 n^{\nu - \frac{1}{2}}\right) \quad (4)$$

*with probability $1 - o(1)$ as $d, n, m \to \infty$. Moreover, if $\nu < \frac{1}{2}$ then with the same probability*

$$\frac{\|S_1\|_2}{\|E\|_2} = \Omega\left(\frac{n^{\frac{1}{2} - \nu - \frac{\alpha}{2}}}{\log n \|W\|_2}\right), \qquad \frac{\|S_2\|_2}{\|E\|_2} = \Omega\left(\frac{n^\nu}{\log n}\frac{\|(z \circ r)^T \sigma'_\perp(XW^T)\|_2}{\|\sigma'_\perp(XW^T)\|_2}\right), \quad (5)$$

$$\frac{\|S_{12}\|_2}{\|E\|_2} = \Omega\left(\frac{n^{\frac{1}{2} - \frac{\beta}{2}}}{\log n \|W\|_2}\right), \qquad \Omega(n^{\nu - \frac{\beta}{2}}) \leq \frac{\|S_{12}\|_2}{\|S_1\|_2} \leq O(n^{\nu - \frac{\beta}{2} + \frac{\alpha}{2}}). \quad (6)$$

Observe that for $\nu < 1/4$, if $\|W\|_2 \log n = o(n^{\frac{1}{2} - \nu - \frac{\alpha}{2}})$ then $G$ is approximately equal to the rank-one matrix $S_1 + S_{12}$. Further, if $\beta > 2\nu + \alpha$ then the gradient is dominated by $S_1$ and the spike is aligned with the data-residue term $X_B^T r$. However, if $\beta < 2\nu$ then the gradient term is dominated by $S_{12}$, which is aligned with the data spike $q$. In addition, for $\nu \in [1/4, 1/2)$, if $\|W\|_2 \log n = o(n^{\frac{1}{2} - \nu - \frac{\alpha}{2}})$ and

$$n^\nu = \omega\left(\log n \frac{\|\sigma'_\perp(XW^T)\|_2}{\|(z \circ r)^T \sigma'_\perp(XW^T)\|_2}\right), \quad (7)$$

then the gradient is approximately the rank-two matrix $S_1 + S_{12} + S_2$. Note this is distinct from prior works [3, 4, 14, 45] where the gradient is only ever approximately rank one.

### 3.2  Small-to-Moderate Spike: ReLU Activation

Theorem 3.1 requires the activation to be $\mathcal{C}^2$. As detailed in the proof, this is needed to establish that $\|\sigma'_\perp(XW^T)\|_2 \leq O(\|W\|n^{\nu + \frac{1}{2}})$. Indeed, when $\|W\|_2 = \Theta(1)$ and $\nu < \frac{1}{2}$ we have $\|\sigma'_\perp(XW^T)\|_2 = o(n)$, which is key for $S_1$ to separate from the bulk spectrum. However, ReLU is not $\mathcal{C}^2$. To understand, the effect of using ReLU we provide Proposition 3.1.

**Proposition 3.1** (ReLU gradient). *If $2\nu > 1 - \alpha$, and the row of $W$ are i.i.d. from the unit sphere, then with probability $1 - o(1)$ we have that $\sigma'_\perp(XW^T) = \frac{1}{2}\,\mathrm{sign}(z_i)\,\mathrm{sign}(Wq)^T$.*

From Proposition 3.1 we see that for ReLU the operator norm of $\sigma'_\perp(XW^T)$ is $\Theta(n)$. This is a significant increase compared to the $o(n)$ scaling for $\mathcal{C}^2$ activations and suggests that the norm of

---

[3]The result [3, Proposition 2] requires $\nu = \alpha = 0$, isotropic data and MSE loss with MF scaling.

$E$ and $S_2$ are larger for ReLU. The increased size of $E, S_2$ results in the relative suppression of the contribution of $S_1$ and an enhancement of the contribution of $S_2$ to the spectrum of the gradient. We empirically verify this phenomenon in Figure 2 where we compare ReLU to its $\mathcal{C}^2$ activations ELU, Swish, and Softplus. We see that, for ReLU the relative residue contribution $(S_1)$ is significantly smaller when compared with its smooth approximations.

**Remark 1** (Convolutional filters inherit the rank-two gradient). A 1D valid convolution with stride 1 and filter $w \in \mathbb{R}^k$ can be written as a two-layer network with a sparse, weight-tied matrix $W \in \mathbb{R}^{m \times d}$ whose nonzeros are shifted copies of $w$, where $m = d - k + 1$. Treating the entries of $W$ as independent parameters yields the gradient $G = \partial L / \partial W \in \mathbb{R}^{m \times d}$. By Theorems 3.1-3.2, $G$ admits a decomposition $G = u^{(1)}(v^{(1)})^\top + u^{(2)}(v^{(2)})^\top + E$ with at most two dominant rank-one terms and a small bulk $E$. Weight tying maps $G$ to the true filter gradient as follows:

$$\frac{\partial L}{\partial w_\ell} = \sum_{i=1}^{m} G_{i,\, i+\ell-1}, \qquad \ell = 1, \ldots, k.$$

Hence, letting $\tilde{v}_i^{(j)} = v_{d-i+1}^{(j)}$ for $j = 1, 2$,

$$\nabla_w L = u^{(1)} * \tilde{v}^{(1)} + u^{(2)} * \tilde{v}^{(2)} + (\text{error}),$$

the convolutional filter gradient lies in a subspace of dimension at most two, upto a small error term.

### 3.3   Large Spike ($\nu \geq 0.5$): Non $\mathcal{C}^2$ Activations and Dependence between $W$ and $X$

The preceding analysis focused on $\nu < 0.5$. For large data spikes ($\nu \geq 0.5$), we note that the $\mathcal{C}^2$ smoothness of the activation function and independence between W and X are no longer required.

**Theorem 3.2** (Large data-spike gradient approximation). *Suppose Assumptions 1, 2, 3, 4, 5, and 6 are satisfied, and define $E_L = G - S_{12} - S_2$. Then, with probability $1 - o(1)$ for $\nu \geq \frac{1}{2}$ we have*

$$\frac{\|E_L\|_2}{\sqrt{m}\gamma_m \|r\|_\infty} = O(1),\; \frac{\|S_{12}\|_2}{\|E_L\|_2} = \Omega\left(\frac{n^{\nu - \frac{\beta}{2}}}{\log n}\right),\; \frac{\|S_2\|_2}{\|E_L\|_2} = \Omega\left(\frac{n^\nu}{\log n} \frac{\|(z \circ r)^T \sigma'_\perp (XW^T)\|_2}{\|\sigma'_\perp (XW^T)\|_2}\right).$$

$$(8)$$

Note this is a generalization of [4], which required alignment between the targets $y$ and the spike $q$. Theorem 3.2 shows that if $\nu > \frac{\beta}{2}$, or if Equation (7) holds, then the gradient is approximately rank one. In contrast to the $\nu < \frac{1}{4}$ case, this rank-one gradient aligns closely with the data spike plus interpolant $S_{12} + S_2$ rather than the residue $S_1$. This is empirically verified in Figure 3 for non-$\mathcal{C}^2$ activations ReLU, as well as dependent and independent $W$ and $X$.

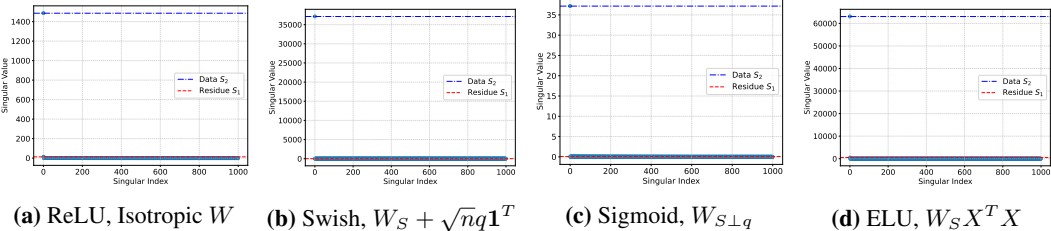

**(a)** ReLU, Isotropic $W$    **(b)** Swish, $W_S + \sqrt{n}q\mathbf{1}^T$    **(c)** Sigmoid, $W_{S \perp q}$    **(d)** ELU, $W_S X^T X$

**Figure 3:** Singular value distributions of the gradient $G$ under various activation functions and weight matrix initializations and structures, with a large data spike $\nu = 3/4$. $W_S$ denotes the random matrix with rows drawn mutually i.i.d. uniformly from the unit sphere. The rows of $W_{S \perp q}$ are uniform on the sphere and orthogonal to $q$. All weight matrices are subsequently normalized to have unit norm rows. Fixed parameters: bulk decay exponent $\alpha = 0$, $n = 750$, $d = 1000$, $m = 1250$, NTK-like scaling ($\gamma_m = 1/\sqrt{m}$), MSE loss, and triple-index model targets.

### 3.4   Impact of the Scale Parameter: MF vs. NTK Scaling

We consider the implications of our results for the two scaling regimes and highlight three important distinctions. As with prior work, we consider the large step-size regime. Specifically, we use a step

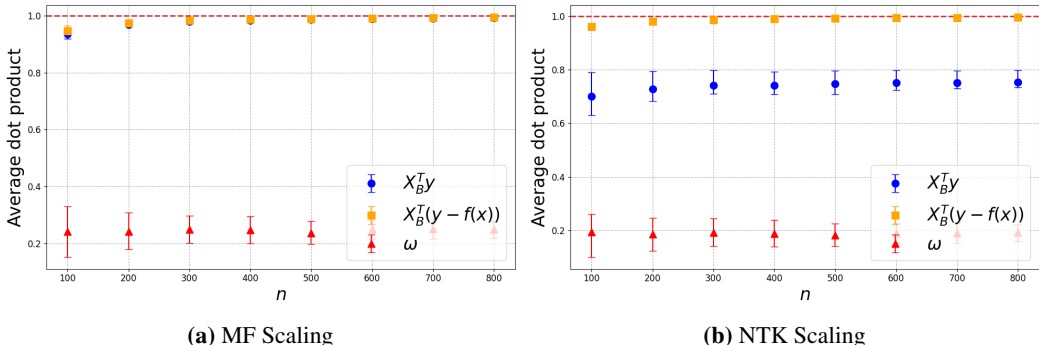

**(a)** MF Scaling        **(b)** NTK Scaling

**Figure 4:** Empirical alignment (normalized inner product) of the top singular vector of the gradient $G$ with $X_B^T y$, $X_B^T r$ and $\omega$ for data from a single-index model $y = \text{Sigmoid}(\omega^T x) + \text{noise}$. We use isotropic $X$, ReLU activation, and MSE loss. We average over 500 samples of $a, W, X, y$. The error bars are the 25th and 75th percentile.

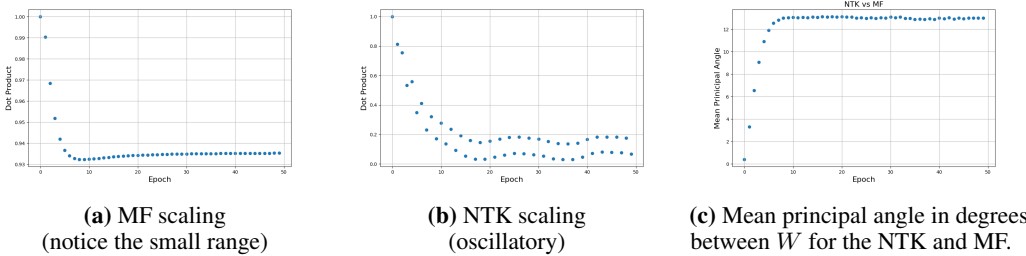

**(a)** MF scaling    **(b)** NTK scaling    **(c)** Mean principal angle in degrees
(notice the small range)    (oscillatory)    between $W$ for the NTK and MF.

**Figure 5:** Evolution of the gradient direction and weight matrix during training under GD with Weight Normalization (WN) for the MF and NTK scalings. Fixed parameters are $\nu = 0, \alpha = 0$ while using the Sigmoid activation function and the MSE loss. Plots (a) and (b) show the alignment (normalized inner product) between the leading left singular vector of the initial gradient $G_0$ (epoch 0) and that of $G_t$ (epoch $t$). Plot (c) shows the mean principal angle between the weight matrices learned under the MF and NTK scalings with identical initialization and training data.

size of $\gamma_m^{-1}$. To avoid exploding gradients deploy *Weight Normalization* (WN) [37]. We limit our focus to the MSE loss. See Appendix D for a discussion of which assumptions hold during training.

**1) Alignment at initialization: residue $r$ versus target vector $y$.** Recall from Theorem 3.1 that in the small spike regime the gradient is dominated by $S_1$. Further, for the MF scaling the residue $r$ is approximately equal to the target $y$, while for the NTK scaling the residue can be quite distinct from $y$. This implies the alignment of the gradient may differ significantly depending on whether an MF or NTK scaling is used. Suppose $y = \text{sigmoid}(\omega^T x) + \varepsilon$, then Figure 4 presents the normalized inner products between the leading left singular vector of $G$ and three candidate directions $X_B^T y$, $X_B^T r$, and $\omega$. For the MF scaling, we see that the gradient's dominant direction aligns well with $X_B^T r, X_B^T y$, consistent with Theorem 3.1 and [3]. For the NTK scaling, consistent with Theorem 3.1, the gradient exhibits strong alignment with $X_B^T r$. This differs notably from both $X_B^T y$ and the $\omega$ alignment directions predicted in [28] which we believe to be erroneous.

**2) Stability of the gradient during early training.** Let $G_t$ denote the gradient after $t$ iterations of GD. In Figure 5 we plot the alignment between the leading left singular vector of $G_0$ and subsequent leading left singular vectors of $G_t$ under both MF and NTK scalings. The following is quite striking: the dominant gradient direction under the MF scaling remains stable throughout training while for the NTK scaling it evolves significantly. This leads to a divergence in the trajectories of the weight matrix even with identical initialization and training data.

Towards explaining this, suppose the conditions of Theorem 3.1 hold at least approximately up to some iteration $t \leq T$. Then under an MF scaling the gradient is approximated by a rank-one matrix whose left singular vector is nearly constant $X_B^T r_t \approx X_B^T y$. Therefore it remains stable over a number of iterations. If the NTK scaling is used instead, then as $S_1$ is proportional to $X_B^T r_t \not\approx X_B^T y$ and the gradient depends on the residuals $r_t$ which evolve throughout training.

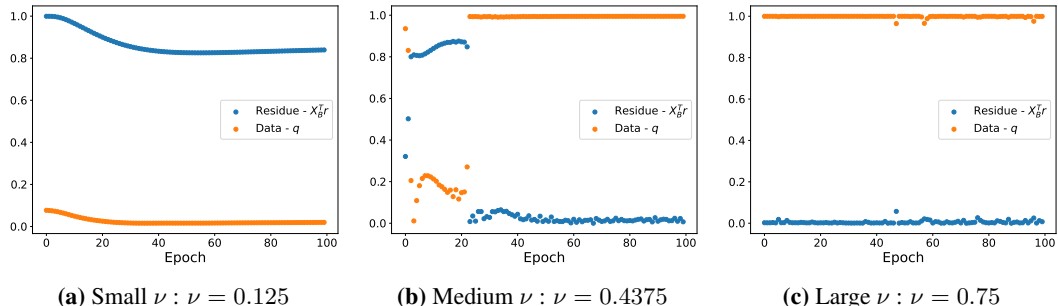

**(a)** Small $\nu$ : $\nu = 0.125$      **(b)** Medium $\nu$ : $\nu = 0.4375$      **(c)** Large $\nu$ : $\nu = 0.75$

**Figure 6:** Evolution of the alignment of the leading left singular vector of $G_t$ with data spike $q$ and residue $(X_B^T r_t)$ during training. Fixed parameters: MF scaling, Tanh activation, MSE loss, $\alpha = 0$.

**3) Phase transitions both by epoch and data spike size.** In Figure 6 we observe the evolution of the alignment of the gradient versus the data spike and the residue under the MF scaling. Moving from small to large spike sizes we observe a transition in the gradient alignment from the residue $X_B^T r$ to the data spike $q$. We remark that this is as predicted by Theorem 3.1 and Theorem 3.2 at initialization. Of particular interest is the middle spike size setting, where we witness a phase transition during training of the gradient alignment from residue to data spike. Interestingly, this transition is not discernible from the training loss, which smoothly decays during training. We only pause to highlight this interesting phenomenon here and leave a more thorough analysis to future work.

## 4 Effect of Regularization

We analyze three common regularization techniques: $\ell_2$ weight decay, isotropic input noise, and Jacobian (or gradient) penalization. We investigate how each technique influences the relative magnitudes of the residue-aligned spike $S_1$ and the data-aligned spike $S_2$. For what follows let $G^{(0)}$ denote the un-regularized gradient matrix derived in Proposition 2.1.

$\ell_2$ **weight decay.** Adding the term $\frac{\lambda}{2}\|W\|_F^2$ to the loss function modifies the gradient to $G^{(\lambda)} = G^{(0)} + \lambda W$. Proposition 4.1 implies that if $\lambda\|W\|_2 = o(\sqrt{m}\gamma_m)$ it cannot suppress $S_1$ or $S_2$, however, if $\lambda\|W\|_2 = \omega(\sqrt{m}\gamma_n n^\nu)$ then it suppresses both spikes.

**Proposition 4.1.** *Given Assumptions 1, 2, 3, 4, and 6. If $\|r\|_2 = O(\sqrt{n})$, then with probability $1-o(1)$ we have that $\|S_1\|_2 \leq O(\sqrt{m}\gamma_m)$, $\|S_{12}\|_2 \leq O(\sqrt{m}\gamma_m n^{\nu-\frac{\beta}{2}})$, and $\|S_2\|_2 \leq O(\sqrt{m}\gamma_m n^\nu)$.*

**Isotropic Gaussian input noise.** This regularization technique involves adding independent isotropic Gaussian noise $\xi_i \sim \mathcal{N}(0, \tau^2 I)$ to each input $x_i$ without changing the corresponding labels $y_i$. [7] showed that training with input noise is equivalent under certain conditions to adding a Tikhonov regularizer to the loss, often related to $\sum_{i=1}^n \|\nabla_x f(x_i)\|_2^2$. More recent work [46] connects adding isotropic noise to the data to controlling the trace of the Hessian of the loss function.

Let us define $x_i' = x_i + \xi_i$. This changes the input data distribution, effectively modifying the bulk covariance from $\hat{\Sigma}$ to $\hat{\Sigma}' = \hat{\Sigma} + \tau^2 I$. Consequently, derived quantities such as the residue vector $r'$, the alignment parameter $\beta'$, the gradient components $S_1', S_2', S_{12}'$, the error term $E'$, and the effective bulk spectral decay $\alpha'$ are denoted with primes.

**Proposition 4.2** (Isotropic Gaussian noise). *Assume the setup of Assumptions 1, 2, 3 with independent $X$ and $W$. Assume $\sigma$ satisfies Assumption 4 for the noisy data $X'$. Additionally, suppose the modified residues satisfy $r_i' = \Theta(1)$ with probability $1 - o(1)$, and Assumption 6 holds for $r'$ with scaling parameter $\beta'$. If $\tau^2 = n^\rho$ and $\|\sigma_\perp'(X'W^T)\|_2 = o(n)$, then with high probability:*

$$\frac{\|S_1'\|_2}{\|E'\|_2} \geq \omega(1), \quad \frac{\|S_2'\|_2}{\|E'\|_2} \leq O(n^{\nu-\frac{\rho}{2}}), \quad \frac{\|S_{12}'\|_2}{\|E'\|_2} \leq o(n^{\nu-\frac{\rho}{2}-\frac{\beta'}{2}}).$$

Proposition 4.2 analyzes the effect of input noise. It indicates that the residue spike $S_1'$ remains prominent relative to the error term $E'$. Conversely, if the noise is sufficiently strong, the data spike components $S_2'$ and $S_{12}'$ become suppressed relative to $E'$. Intuitively, adding noise with variance $\tau^2 = n^\rho$ increases the variance of the bulk data component. This boosts the overall scale of terms involving $(X_B')^T$. Simultaneously, the added noise tends to make the pre-activations $W^T X'$

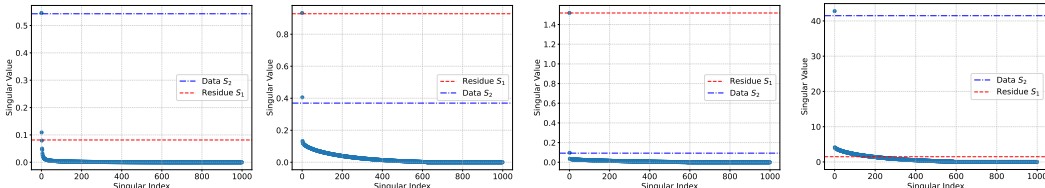

(a) $\tau^2 = 0$. ReLU suppresses the residue spike.

(b) $\tau^2 = 0.25$. The residue spike re-appears.

(c) $\lambda = 0$. Residue spike is dominant.

(d) $\lambda = 100$. Data spike is dominant.

**Figure 7:** Effect of regularization. Panels **(a)**, **(b)** are for isotropic Gaussian noise. Parameters: $n = 750, d = 1000, m = 1250, \nu = 1/8, \alpha = 8/9$ (for original data), triple-index targets, ReLU activation, MSE loss. Panels **(c)**, **(d)** are for Jacobian norm penalization. As $\lambda$ increases, the size of $S_1$ does change, size of the bulk $E_2$ grows, and the size of the data spike $S_3$ grows. Parameters: NTK, $\nu = 3/8, \alpha = 0$, Sigmoid and MSE, and triple-index model targets.

more isotropic, which can reduce the operator norm $\|\sigma'_\perp(X'W^T)\|_2$ relative to its Frobenius norm, potentially limiting the growth rate of $\|E'\|_2, \|S'_2\|$. This predicted relative enhancement of $S'_1$ and suppression of $S'_2$ is verified empirically. As discussed in Section 3 (cf. Proposition 3.1), ReLU can hinder residue spike $S_1$. However, Figure 7 shows that with small amount of input noise $\tau^2 = 0.25$, an initially suppressed $S'_1$ re-emerges, while $S'_2$ is diminished relative to $S'_1$ and the bulk.

**Jacobian penalization.** Another form of regularization penalizes the sensitivity of the network output to changes in the inner weights. We consider the Jacobian penalty $L_{reg} = \lambda \frac{1}{2n} \sum_{i=1}^{n} \|\partial_W f(x_i)\|_2^2$. To analyze this effect of $L_{reg}$ on the gradient, we derive the gradient of $L_{reg}$ with respect to $W$.

**Proposition 4.3** (Gradient penalty). *Let* $\mathrm{Diag}(\|x_i\|^2)$ *be the* $n \times n$ *diagonal matrix, whose entries are* $\|x_i\|^2$. *If* $\sigma$ *is twice differentiable, then*

$$\nabla_W L_{reg} = \frac{1}{n} \lambda \gamma_m^2 \left( \sigma'(WX^T) \odot \sigma''(WX^T) \right) \mathrm{Diag}(\|x_i\|^2) X.$$

The gradient of the regularizer factorizes into a *data-aligned* rank-one spike $S_3$ and error $E_2$:

$$S_3 = \frac{1}{n} \gamma_m^2 X_S^T \Psi, \quad E_2 = \frac{1}{n} \gamma_m^2 X_B^T \Psi, \quad \Psi = \mathrm{Diag}(\|x_i\|^2) \left( \sigma'(XW^T) \odot \sigma''(XW^T) \right).$$

**Proposition 4.4.** *Given Assumptions 1, 2, 3, 4, and 6. If* $\|r\|_2 = \Theta(\sqrt{n})$, $\alpha < 1$, *and a constant fraction of the entries of* $\sigma'(XW^T) \odot \sigma''(XW^T)$ *are bounded away from 0, then*

$$\lambda \left( n^{2\nu - \frac{\alpha}{2} - \frac{1}{2}} + n^{\frac{1-3\alpha}{2}} \right) \geq \sqrt{m} \gamma_m \frac{\|\lambda E_2\|_2}{\|S_1\|_2} \geq \lambda \left( n^{2\nu - \frac{\alpha}{2} - 1} + n^{-\frac{3}{2}\alpha} \right).$$

If $\nu > \frac{1}{2} + \frac{\alpha}{2}$, then we have that asymptotically the residue spike does not escape the bulk for any $\lambda = \Theta(1)$. If $\nu < \frac{1}{2}$, we see that increasing $\lambda$ suppresses the residue spike. For the data spike, we have that $\lambda S_3$ will grow as $\lambda$ grows. Hence this enhances the data spike. We empirically verify that increasing $\lambda$ kills the residue spike while promoting the data spike (Figure 7).

**Real-Data validation.** The identified low-rank spike-plus-bulk gradient structure and the discussed regularization effects are observable in two standard vision datasets - MNIST and CIFAR10. For MNIST, we estimate $\nu \approx 0.784 > 1/2$ and the data is highly ill-conditioned, suggesting a large effective $\alpha$. Theorem 3.2 predicts a gradient dominated by data-aligned components (Panel **(c)** of Figure 8). Adding isotropic Gaussian noise with $\sigma^2 = 100$ (Panel **(d)**) suppresses the original data-aligned spike and enhances the residue-aligned spike $S_1$, consistent with the analysis in Section 4. For CIFAR-10 we use a pretrained ResNet-18 (on ImageNet) to extract 512-dimensional embedding. We estimate $\nu \approx 0.3572 < 1/2$ and $\alpha \approx 0.6$. For these parameters Theorem 3.1 suggests $S_1$ (residue-aligned) can be prominent. Panel **(a)** of Figure 8 shows a dominant $S_1$. Applying Jacobian regularization with $\lambda = 10^5$ (Panel **(b)**) suppresses $S_1$ and promotes a data-aligned spike (akin to $S_2$), consistent with the behavior analyzed for Jacobian penalization in Section 4.

## 5 Conclusion

This work shows that in two-layer neural networks, the hidden-layer gradient is approximately rank-two, driven by data-residual ($S_1$) and data-spike ($S_2$) components connected by an interpolant

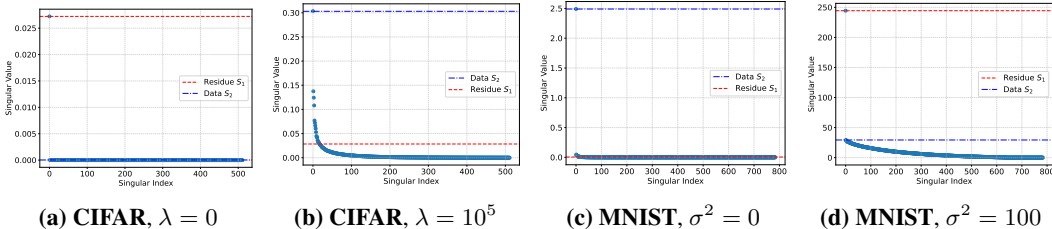

**(a) CIFAR**, $\lambda = 0$     **(b) CIFAR**, $\lambda = 10^5$     **(c) MNIST**, $\sigma^2 = 0$     **(d) MNIST**, $\sigma^2 = 100$

**Figure 8:** Gradient singular value spectra on real datasets. Each panel displays the singular values of the gradient matrix $G$ under the specified conditions.

($S_{12}$). We show that activation function choice, scaling, and regularization can result in qualitatively different gradients. In particular, we have the following rule of thumb for the number of spikes.

---

Gradient-spike rule-of-thumb: Which spike dominates at initialization?

$$\boxed{S_1 \text{ (residue spike)}} \;\leftrightarrow\; 2\nu < \min\{\tfrac{1}{2}, \beta - \alpha, 1 - \alpha\} \text{ or Large isotropic input noise}$$

$$\boxed{S_{12} + S_2 + S_3 \text{ (data spike)}} \;\leftrightarrow\; \begin{cases} (i)\ 2\nu > \min\{1, \beta\}, \text{ or } (ii)\ \text{Strong Jacobian penalty,} \\ \text{or } (iii)\ \text{ReLU and } 2\nu > 1 - \alpha \end{cases}$$

If none of the above holds, both spikes remain, and the gradient is typically rank-two.

---

The coexistence and interplay of the two spike components offer a nuanced understanding of the gradient. We believe that the residue-aligned part propels the network towards fitting the current errors for the specific task, while the data-aligned part reflects the network's adaptation to or influence by the inherent structure and biases present in the input data distribution. This dual influence provides a potential mechanism for reconciling how networks can be both task-specific and data-adaptive. This is an interesting avenue for future work.

### Acknowledgments

GM has been supported in part by the NSF in NSF CCF-2212520 and NSF DMS-2145630. GM has also been supported by DARPA in AIQ project HR00112520014, DFG in SPP 2298 (FoDL) project 464109215, and BMFTR in DAAD project 57616814 (SECAI).

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

# Contents

**Table 1:** Notation

| Symbol | Meaning | Where first defined / used |
| --- | --- | --- |
| $d$ | Input dimension | Assumption 1 |
| $n$ | Number of samples | Assumption 1 |
| $m$ | Hidden-layer width | Assumption 1 |
| $\psi_1 = n/d, \ \psi_2 = m/d$ | Proportional-scaling ratios | Assumption 1 |
| $\widehat{\Sigma}$ | Bulk covariance matrix | Assumption 2 |
| $\Sigma = \widehat{\Sigma} + \zeta^2 q q^\top$ | Full covariance (bulk + spike) | Assumption 2 |
| $\lambda_k = k^{-\alpha}$ | Bulk eigen-spectrum | Assumption 2 |
| $\alpha \geq 0$ | Spectral-decay exponent | Assumption 2 |
| $\zeta = n^\nu, \ \nu \geq 0$ | Spike magnitude | Assumption 2 |
| $q \in \mathbb{S}^{d-1}$ | Spike direction | Assumption 2 |
| $z \in \mathbb{R}^n$ | Latent coordinates of the spike | Equation (3) |
| $X = X_B + X_S$ | Data split bulk + spike | Equation (3) |
| $X_B$ | Bulk part ($\mathcal{N}(0, \widehat{\Sigma})$ rows) | Equation (3) |
| $X_S = \zeta z q^\top$ | Rank-1 spike part | Equation (3) |
| $W \in \mathbb{R}^{m \times d}$ | Inner-layer weight matrix | Assumption 3 |
| $a \in \{\pm 1\}^m$ | Outer weights (fixed) | Assumption 3 |
| $\gamma_m$ | Width scale (NTK $= 1/\sqrt{m}$, MF $= 1/m$) | Equation (1) |
| $\sigma, \sigma', \sigma''$ | Activation and derivatives | Assumption 4 |
| $\mu = \mathbb{E}_x[\sigma'(Wx)]$ | Mean derivative vector | Assumption 4 |
| $\sigma'_\perp = \sigma' - \mu$ | Centered derivative | Assumption 4 |
| $r$ | Residue vector | Equation (2) |
| $\beta$ | Alignment exponent ($\frac{1}{\sqrt{n}\,\|r\|_2} z^\top r$) | Assumption 6 |
| $S_1$ | Residue-aligned rank-1 term | Section 3 |
| $S_2$ | Data-spike-aligned rank-1 term | Section 3 |
| $S_{12}$ | Interpolant rank-1 term | Section 3 |
| $G = \nabla_W \mathcal{L}$ | Full gradient wrt $W$ | Prop. 2.1 |
| $E$ | Error term $G - S_1 - S_{12} - S_2$ | Thm. 3.1 |
| $E_L$ | Error term $G - S_{12} - S_2$ (large-spike version) | Thm. 3.2 |
| $E_2$ | Bulk error from Jacobian-penalty gradient | Prop. 4.3 |
| $S_3$ | Data-aligned rank-1 term induced by Jacobian penalty | Prop. 4.3 |
| $\lambda, L_{\text{reg}}$ | Reg. strength and Jacobian penalty | Section 4 |
| $\tau^2$ | Variance of isotropic Gaussian noise | Section 4 |
| $\circ, \otimes$ | Hadamard / outer products | |

## A  Proofs

**Notation** In the appendix, we shall use $f \lesssim g$ to mean that $f = O(g)$ with probability $1 - o(1)$.

### A.1  Regularization Proofs

**Proposition 4.1.** *Given Assumptions 1, 2, 3, 4, and 6. If $\|r\|_2 = O(\sqrt{n})$, then with probability $1 - o(1)$ we have that $\|S_1\|_2 \leq O(\sqrt{m}\gamma_m)$, $\|S_{12}\|_2 \leq O(\sqrt{m}\gamma_m n^{-\frac{\beta}{2}})$, and $\|S_2\|_2 \leq O(\sqrt{m}\gamma_m n^\nu)$.*

*Proof.* These bound immediately follow from Lemma A.1, Lemma A.2, and Lemma A.3. □

**Proposition 4.2** (Isotropic Gaussian noise)**.** *Assume the setup of Assumptions 1, 2, 3 with independent $X$ and $W$. Assume $\sigma$ satisfies Assumption 4 for the noisy data $X'$. Additionally, suppose the modified residues satisfy $r'_i = \Theta(1)$ with probability $1 - o(1)$, and Assumption 6 holds for $r'$ with scaling parameter $\beta'$. If $\tau^2 = n^\rho$ and $\|\sigma'_\perp(X'W^T)\|_2 = o(n)$, then with high probability:*

$$\frac{\|S'_1\|_2}{\|E'\|_2} \geq \omega(1), \quad \frac{\|S'_2\|_2}{\|E'\|_2} \leq O(n^{\nu - \frac{\rho}{2}}), \quad \frac{\|S'_{12}\|_2}{\|E'\|_2} \leq o(n^{\nu - \frac{\rho}{2} - \frac{\beta'}{2}}).$$

*Proof.* We prove each bound in turn.

$S'_1$ **Bound:**  Recall that $S'_1 = \frac{\gamma_m}{n}(X'_B)^T r'(a \circ \mu')^T$. Since $d > n$, and $X'_B \in \mathbb{R}^{n \times d}$ is full rank with probability 1, we have that with probability 1, for any vector $v$

$$\|(X'_B)^T v\|_2 \geq \sigma_{min}(X)\|v\|_2$$

Since the smallest eigenvalue of $\hat{\Sigma}'$ is $n^\rho$, with probability $1 - o(1)$, we have that

$$\sigma_{min}(X'_B) \geq n^{\frac{1}{2} + \frac{\rho}{2}}.$$

Applying to $S'_1$, we get

$$\|S'_1\|_2 \gtrsim \gamma_m n^{\rho/2} \|a \circ \mu'\|_2 \frac{\|r\|_2}{\sqrt{n}}$$

Then using Assumption 4, the fact that the entries of $a$ are $\pm 1$, and $r'_i = \Theta(1)$, we get

$$\|S'_1\|_2 \gtrsim \gamma_m n^{\rho/2} \sqrt{m}$$

$E$ **Bound:** Next, we have that $E' = \frac{\gamma_m}{n}(X'_B)^T((r'a^T) \circ \sigma'_\perp(X'W^T))$. Using the fact that with probability $1 - o(1)$, $r'_i = \Theta(1)$ and $a_i = \pm 1$, we have that with probability $1 - o(1)$

$$\|(r'a^T) \circ \sigma'_\perp(X'W^T)\|_2 = \|\sigma'_\perp(X'W^T)\|_2.$$

Thus, we have that

$$\|E'\|_2 \lesssim \frac{\gamma_m}{n} \|X'_B\|_2 \|\sigma'_\perp(X'W^T)\|_2$$
$$\lesssim \frac{\gamma_m}{n} \sqrt{n} n^{\rho/2} \|\sigma'_\perp(X'W^T)\|_2$$

Since $d > n$ and $X'_B$ if full rank with probability 1, we have that with probability 1,

$$\|(X'_B)^T((r'a^T) \circ \sigma'_\perp(X'W^T))\|_2 \geq \sigma_{min}(X'_B)\|((r'a^T) \circ \sigma'_\perp(X'W^T))\|_2$$
$$= \sigma_{min}(X'_B)\|\sigma'_\perp(X'W^T))\|_2$$

Hence we get

$$\|E'\|_2 \gtrsim \frac{\gamma_m}{n} \sqrt{n} n^{\rho/2} \|\sigma'_\perp(X'W^T)\|_2$$

$S'_2$ **Bound:** Recall that

$$S'_2 = \frac{\gamma_m}{n} n^\nu q z^T((r'a^T) \circ \sigma'_\perp(X'W^T))$$

Hence we get that

$$\|S'_2\|_2 = \frac{\gamma_m}{n} n^\nu \|q\|_2 \|z^T((r'a^T) \circ \sigma'_\perp(X'W^T))\|_2$$
$$\leq \frac{\gamma_m}{n} n^\nu \|z\|_2 \|((r'a^T) \circ \sigma'_\perp(X'W^T))\|_2$$
$$\lesssim \frac{\gamma_m}{n} n^{\nu + \frac{1}{2}} \|\sigma'_\perp(X'W^T))\|_2$$

$S'_{12}$ **Bound:** Recall that

$$S'_{12} = \frac{\gamma_m}{n} n^{nu} q z^T r'(a \circ \mu')^T$$

Thus, we have that

$$\|S'_{12}\|_2 = \frac{\gamma_m}{n} n^{nu} \|z^T r(a \circ \mu')\|_2$$
$$= \frac{\gamma_m}{n} n^{nu} \|z^T r\|_2 \|(a \circ \mu')\|_2$$
$$\lesssim \frac{\gamma_m}{n} n^{nu - \beta'/2} \|r'\|_2 \|z\|_2 \|(a \circ \mu')\|_2$$
$$\lesssim \sqrt{m} \gamma_m n^{\nu - \frac{\beta'}{2}}$$

**Relative Bounds:** Thus, we have that using $\|\sigma'_\perp(X'W^T)\|_2 = o(n)$

$$\frac{\|S'_1\|_2}{\|E'\|_2} \gtrsim \frac{n}{\|\sigma'_\perp(X'W^T)\|_2} = \omega(1)$$

For the upper bounds we see that

$$\frac{\|S'_2\|_2}{\|E'\|_2} \lesssim n^{\nu-\frac{\rho}{2}}, \quad \frac{\|S'_{12}\|_2}{\|E'\|_2} \lesssim n^{\nu-\frac{\beta'}{2}-\frac{\rho}{2}} \cdot \frac{\|\sigma'_\perp(X'W^T)\|_2}{n} = n^{\nu-\frac{\beta'}{2}-\frac{\rho}{2}} o(1).$$

$\square$

**Proposition 4.3** (Gradient penalty). *Let* $\mathrm{Diag}(\|x_i\|^2)$ *be the* $n \times n$ *diagonal matrix, whose entries are* $\|x_i\|^2$. *If* $\sigma$ *is twice differentiable, then*

$$\nabla_W L_{reg} = \frac{1}{n}\lambda\gamma_m^2 \left(\sigma'(WX^T) \odot \sigma''(WX^T)\right) \mathrm{Diag}(\|x_i\|^2)X.$$

*Proof.* Letting $Z = WX^T$ and $f_i = f(x_i)$ then note

$$f_i = a^T\sigma(Wx_i) = a^T h_i, \text{ and } \partial_{h_i} f_i = a.$$

It follows that

$$\partial_{z_i} f_i = \partial_{h_i} f_i \odot \sigma'(z_i) = a \odot \sigma'(z_i).$$

Recall

$$\frac{\partial Z_{rc}}{\partial W_{kj}} = \mathbf{1}_{\{c=k\}} X_{rj},$$

then

$$\frac{\partial f_i}{\partial W_{kj}} = \sum_{c=1}^m \frac{\partial f_i}{\partial Z_{ic}} \frac{\partial Z_{ic}}{\partial W_{kj}} = \frac{\partial f_i}{Z_{ik}} X_{ij}$$

and therefore

$$\partial_W f_i = (a \odot \sigma'(Wx_i))x_i^T$$

Let $g_i = a \odot \sigma'(h_i)$, then

$$\|\partial_W f_i\|_F^2 = \|g_i x_i^T\|_F^2 = \sum_{j,k} g_{ij}^2 x_{ik}^2 = \|g_i\|_2^2 \|x_i\|_2^2.$$

Now

$$\frac{\partial}{\partial W_{rc}}\|g_i\|_2^2 = \frac{\partial}{\partial W_{rc}}\sum_{j=1}^n a_j^2 \frac{\partial}{\partial W_{rc}}\sigma'(w_j^T x_i)^2$$

$$= \left(2a_r^2\sigma'(w_j^T x_i)\sigma''(w_j^T x_i)\right) x_{ic}.$$

The term inside the brackets is independent of $c$ while the term outside the brackets is independent of $r$. As a result this is an outer product and

$$\partial_W\|g_i\|_2^2 = 2\left(a^{\circ 2} \odot \sigma'(Wx_i) \odot \sigma''(Wx_i)\right) x_i^T.$$

Note above $a^{\circ 2}$ refers squaring operation being applied elementwise to the vector $a$. Therefore

$$\partial_W R = \frac{1}{2}\sum_{i=1}^n \partial_W\|\partial_W f_i\|_F^2 \tag{9}$$

$$= \frac{1}{2}\sum_{i=1}^n \|x_i\|_2^2 \partial_W\|g_i\|_2^2 \tag{10}$$

$$= \sum_{i=1}^n \|x_i\|_2^2 \left(a^{\circ 2} \odot \sigma'(Wx_i) \odot \sigma''(Wx_i)\right) x_i^T. \tag{11}$$

$$= \left(a^{\circ 2}\mathbf{1}^T \circ \sigma'(WX^T) \circ \sigma''(WX^T)\right) \mathrm{Diag}(\|x_i\|^2)X \tag{12}$$

$$= \left(\sigma'(WX^T) \circ \sigma''(WX^T)\right) \mathrm{Diag}(\|x_i\|^2)X \tag{13}$$

$\square$

**Proposition 4.4.** *Given Assumptions 1, 2, 3, 4, and 6. If $\|r\|_2 = \Theta(\sqrt{n})$, $\alpha < 1$, and a constant fraction of the entries of $\sigma'(XW^T) \odot \sigma''(XW^T)$ are bounded away from 0, then*

$$\lambda\left(n^{2\nu-\frac{\alpha}{2}-\frac{1}{2}} + n^{\frac{1-3\alpha}{2}}\right) \geq \sqrt{m}\gamma_m \frac{\|\lambda E_2\|_2}{\|S_1\|_2} \geq \lambda\left(n^{2\nu-\frac{\alpha}{2}-1} + n^{-\frac{3}{2}\alpha}\right).$$

*Proof.* We begin by noting that since $\sigma, \sigma'$ are lipschitz, we have that $\sigma', \sigma''$ are bounded. Hence

$$\sigma'(XW^T) \odot \sigma''(XW^T)$$

has an operator norm that is at most $O(n)$. Since a constant fraction $p$ of the entries are at least some universal constant $c$, then in the proportional regime, we have that

$$\|\sigma'(XW^T) \odot \sigma''(XW^T)\|_2 \geq \frac{1}{\sqrt{n}}\|\sigma'(XW^T) \odot \sigma''(XW^T)\|_F \gtrsim \sqrt{m}c = \Omega(\sqrt{n})$$

Recall that

$$E_2 = \frac{1}{n}\gamma_m^2 X_B^T \operatorname{Diag}(\|x_i\|^2)\left(\sigma'(XW^T) \odot \sigma''(XW^T)\right).$$

Then since $d > n$, $X_B^T \operatorname{Diag}(\|x_i\|^2)$ is full rank with probability 1, we have that

$$\frac{1}{n}\gamma_m^2 \sigma_{min}\left(X_B^T \operatorname{Diag}(\|x_i\|^2)\right)\|\sigma'(XW^T) \odot \sigma''(XW^T)\|_2 \lesssim \|E_2\|_2$$

and

$$\|E_2\|_2 \lesssim \frac{1}{n}\gamma_m^2 \sigma_{max}\left(X_B^T \operatorname{Diag}(\|x_i\|^2)\right)\|\sigma'(XW^T) \odot \sigma''(XW^T)\|_2$$

Due to Assumption 2, with high probability $1 - o(1)$, we have that

$$\sigma_{max}(X_B) \lesssim \sqrt{n} \text{ and } \sigma_{min}(X_B) \gtrsim n^{\frac{1-\alpha}{2}}$$

Then since $\|x_i\|^2$ concentrates to $n^{2\nu} + n^{1-\alpha}$ (for $\alpha < 1$), we have that

$$\|E_2\|_2 \lesssim \frac{\gamma_m^2}{n}\sqrt{n}(n^{2\nu} + n^{1-\alpha})\|\sigma'(XW^T) \odot \sigma''(XW^T)\|_2$$

and

$$\|E_2\|_2 \gtrsim \frac{\gamma_m^2}{n}n^{\frac{1-\alpha}{2}}(n^{2\nu} + n^{1-\alpha})\|\sigma'(XW^T) \odot \sigma''(XW^T)\|_2$$

Then using the $O(n)$ upper bound on $\|\sigma'(XW^T) \odot \sigma''(XW^T)\|_2$, in the proportional regime, with high probability $1 - o(1)$, we get that

$$\|E_2\|_2 \lesssim m\gamma_m^2(n^{2\nu-\frac{1}{2}} + n^{\frac{1}{2}-\alpha})$$

Using our $\Omega(\sqrt{n})$ lower bound on $\|\sigma'(XW^T) \odot \sigma''(XW^T)\|_2$, we get

$$\|E_2\|_2 \gtrsim m\gamma_m^2(n^{2\nu-\frac{\alpha}{2}-1} + n^{-\frac{3\alpha}{2}})$$

On the other hand, if $\|r\|_2 = \Theta(\sqrt{n})$ we have that

$$\sqrt{m}\gamma_m n^{-\frac{\alpha}{2}} \lesssim \|S_1\|_2 \lesssim \sqrt{m}\gamma_m$$

For the NTK regime, we have that

$$n^{2\nu-\frac{\alpha}{2}-\frac{1}{2}} + n^{\frac{1-3\alpha}{2}} \gtrsim \sqrt{m}\gamma_m \frac{\|E_2\|_2}{\|S_1\|_2} \gtrsim n^{2\nu-\frac{\alpha}{2}-1} + n^{-\frac{3}{2}\alpha}$$

$\square$

## A.2 Spikey Gradient Proof

**Proposition 2.1** (Gradient of the loss). *If Assumption 4 holds and $R$ is differentiable, then*

$$G := \nabla_{W^T} L = \gamma_m X^T \left[ (ra^T) \circ \sigma'(XW^T) \right] + \lambda \nabla_{W^T} R(W) \in \mathbb{R}^{d \times m}$$

*exists for almost every $W$ in $\mathbb{R}^{m \times d}$.*

*Proof.* The first thing we need to do is to compute the gradient. To begin, we compute

$$f(x_i) = \sum_{j=1}^{m} a_j \sigma \left( \sum_{k=1}^{d} w_{jk}(x_i)_k \right)$$

Thus, we see that

$$
\begin{aligned}
\frac{\partial}{\partial w_{rs}} L(f(x)) &= \frac{1}{n} \sum_{i=1}^{n} \ell'(f(x_i)) \frac{\partial}{\partial w_{rs}} \left( \sum_{j=1}^{m} a_j \sigma \left( \sum_{k=1}^{d} w_{jk}(x_i)_k \right) \right) \\
&= \frac{1}{n} \sum_{i=1}^{n} \ell'(f(x_i)) \sum_{j=1}^{m} a_j \frac{\partial}{\partial w_{rs}} \left( \sigma \left( \sum_{k=1}^{d} w_{jk}(x_i)_k \right) \right) \\
&= \frac{1}{n} \sum_{i=1}^{n} \ell'(f(x_i)) \sum_{j=1}^{m} a_j \sigma' \left( w_j^T x_i \right) \frac{\partial}{\partial w_{rs}} \left( \sum_{k=1}^{d} w_{jk}(x_i)_k \right) \\
&= \frac{1}{n} \sum_{i=1}^{n} \ell'(f(x_i)) a_r \sigma' \left( w_r^T x_i \right) (x_i)_s \\
&= \frac{1}{n} \sum_{i=1}^{n} (L'(f(X))a)_{ir} \sigma'(XW^T)_{ir} X_{is} \\
&= \frac{1}{n} (X^T [(L'(f(X))a) \circ \sigma'(XW^T)])_{sr}
\end{aligned}
$$

$\square$

We begin by decomposing the gradient

$$G = \frac{\gamma_m}{n} X^T \left( (ra^T) \circ \sigma'(XW^T) \right).$$

This algebraic decomposition holds for the current state $(X, W, r, a)$, irrespective of any statistical dependence between $W$ and $X$. Recall the data decomposition

$$X = X_B + X_S = X_B + \zeta z q^T \in \mathbb{R}^{n \times d}$$

where rows of $X_B$ are from $\mathcal{N}(0, \hat{\Sigma})$, $z \sim \mathcal{N}(0, I)$, $\|q\| = 1$ and the activation derivative decomposition $\sigma'(XW^T) = \mathbf{1}_n \mu^T + \sigma'_\perp(XW^T)$, where $\mu = \mathbb{E}_x[\sigma'(Wx)]$ depends on the current $W$.

Substituting these into the gradient expression yields:

$$
\begin{aligned}
G &= \frac{\gamma_m}{n} X^T \left( (ra^T) \circ \left[ \mathbf{1}_n \mu^T + \sigma'_\perp(XW^T) \right] \right) \\
&= \frac{\gamma_m}{n} X^T \left( r(a \circ \mu)^T + (ra^T) \circ \sigma'_\perp(XW^T) \right) \\
&= \frac{\gamma_m}{n} (X_B^T + X_S^T) \left( r(a \circ \mu)^T \right) + \frac{\gamma_m}{n} (X_B^T + X_S^T) \left( (ra^T) \circ \sigma'_\perp(XW^T) \right) \\
&= \underbrace{\frac{\gamma_m}{n} X_B^T r(a \circ \mu)^T}_{S_1} + \underbrace{\frac{\gamma_m}{n} X_S^T r(a \circ \mu)^T}_{S_{12}} \\
&\quad + \underbrace{\frac{\gamma_m}{n} X_S^T ((ra^T) \circ \sigma'_\perp(XW^T))}_{S_2} + \underbrace{\frac{\gamma_m}{n} X_B^T ((ra^T) \circ \sigma'_\perp(XW^T))}_{E}.
\end{aligned}
$$

Using $X_S = \zeta z q^T$, we identify the components explicitly:

$$S_1 = \gamma_m \frac{X_B^T r}{n} (a \circ \mu)^T$$

$$S_{12} = \gamma_m \zeta \left( \frac{z^T r}{n} \right) q(a \circ \mu)^T$$

$$S_2 = \frac{\gamma_m \zeta}{n} q \left( z^T ((ra^T) \circ \sigma'_\perp (XW^T)) \right)$$

$$E = \frac{\gamma_m}{n} X_B^T ((ra^T) \circ \sigma'_\perp (XW^T)).$$

Note that $S_{12}$ shares its right singular vector $(a \circ \mu)$ with $S_1$ (up to scaling) and its left singular vector $q$ with $S_2$. Understanding the gradient structure requires bounding the norms of these terms, which depends on the properties of the current $W, r, \mu$, and the data statistics.

### A.2.1 Upper and Lower Bounds

Given our helper results, we now provide bounds for the $S_1, S_{12}, S_2$, and $E$ appearing in Section 3.

**Lemma A.1** ($S_1$ Bound). *Let $W$ be the weight matrix (e.g., at step $t$) with unit norm rows, and let $S_1 = \gamma_m \frac{X_B^T r}{n} (a \circ \mu)^T$. Suppose $X_B$ is from Assumption 2, $a$ has fixed $\pm 1$ entries (Assumption 3), $r$ is the current residual, and $\mu = \mathbb{E}_x[\sigma'(Wx)]$ satisfies $\mu_k = \Theta(1)$ for all $k$ (Assumption 4). Assume $d > n$. Then with high probability:*

$$\sqrt{m} \gamma_m \mu_{\min} \|r\|_2 n^{-\frac{\alpha+1}{2}} \lesssim \|S_1\|_2 \lesssim \sqrt{m} \gamma_m \mu_{\max} \|r\|_2 n^{-\frac{1}{2}},$$

*where $\mu_{\min} = \min_k |\mu_k| = \Omega(1)$ and $\mu_{\max} = \max_k |\mu_k| = O(1)$.*

*Proof.* The operator norm is

$$\|S_1\|_2 = \frac{\gamma_m}{n} \|X_B^T r\|_2 \|a \circ \mu\|_2.$$

First, consider $a \circ \mu$, where $a_k = \pm 1$ and $\mu_k = \mathbb{E}_x[\sigma'(w_k^T x)]$. By assumption, $\mu_{\min} = \min_k |\mu_k| = \Omega(1)$ and $\mu_{\max} = \max_k |\mu_k| = O(1)$ (since $\sigma'$ is bounded). We have:

$$\|a \circ \mu\|_2^2 = \sum_{k=1}^m a_k^2 \mu_k^2 = \sum_{k=1}^m \mu_k^2.$$

Thus, we see that

$$\mu_{\min} \sqrt{m} \leq \|a \circ \mu\|_2 \leq \mu_{\max} \sqrt{m}.$$

By Assumption 2, if $d > n$ we have that with high probability

$$n^{\frac{1-\alpha}{2}} \|r\|_2 \lesssim \|X_B^T r\|_2 \lesssim n^{\frac{1}{2}} \|r\|_2.$$

Substituting the bounds for $\|X_B^T r\|_2$ and $\|a \circ \mu\|_2$ into the expression for $\|S_1\|_2 = \frac{\gamma_m}{n} \|X_B^T r\|_2 \|a \circ \mu\|_2$:

Lower: $\quad \|S_1\|_2 \gtrsim \frac{\gamma_m}{n} (n^{\frac{1-\alpha}{2}} \|r\|_2)(\sqrt{m}\mu_{\min}) = \gamma_m \sqrt{m} \mu_{\min} \|r\|_2 n^{-\frac{\alpha+1}{2}}$

Upper: $\quad \|S_1\|_2 \lesssim \frac{\gamma_m}{n} (n^{\frac{1}{2}} \|r\|_2)(\sqrt{m}\mu_{\max}) = \gamma_m \sqrt{m} \mu_{\max} \|r\|_2 n^{-\frac{1}{2}}.$

This completes the proof. $\qquad \square$

**Lemma A.2** ($S_{12}$ Bound). *Let $W$ be the weight matrix (e.g., at step $t$) with unit norm rows. Let $S_{12} = \gamma_m \zeta (\frac{z^T r}{n}) q(a \circ \mu)^T$. Suppose $z, q, \zeta = n^\nu$ are from Assumption 2, $a$ has fixed $\pm 1$ entries (Assumption 3), $\mu = \mathbb{E}_x[\sigma'(Wx)]$ satisfies $\mu_k = \Theta(1)$ (Assumption 4), and the current residual $r$ satisfies $|\frac{z^T r}{n}| = \Theta(\|r\|_2 n^{-\beta/2-1/2})$ (Assumption 6). Assume $d > n$. Then w.h.p.:*

$$\|S_{12}\|_2 = \Theta \left( \sqrt{m} \gamma_m \|r\|_2 n^{\nu - \frac{\beta}{2} - \frac{1}{2}} \right).$$

*Proof.* Since $S_{12}$ is a rank-1 matrix and $\|q\|_2 = 1$, its operator norm is:

$$\|S_{12}\|_2 = \left|\gamma_m\zeta\left(\frac{z^T r}{n}\right)\right|\|q\|_2\|a \circ \mu\|_2 = \gamma_m n^\nu \left|\frac{z^T r}{n}\right|\|a \circ \mu\|_2.$$

By Assumption 6 applied to the current residual $r$, we have

$$\left|\frac{z^T r}{n}\right| = \Theta\left(\|r\|_2 n^{-\frac{\beta}{2}-\frac{1}{2}}\right).$$

Substituting this scaling, we get

$$\|S_{12}\|_2 = \gamma_m n^\nu \Theta\left(\|r\|_2 n^{-\frac{\beta}{2}-\frac{1}{2}}\right)\|a \circ \mu\|_2 = \Theta\left(\gamma_m n^{\nu-\frac{\beta+1}{2}}\|r\|_2\|a \circ \mu\|_2\right).$$

As established in the proof of Lemma A.1, using the assumptions on $a$ and $\mu$ (specifically $\mu_k = \Theta(1)$), we have $\|a \circ \mu\|_2 = \Theta(\sqrt{m})$. Combining these gives the final result:

$$\|S_{12}\|_2 = \Theta\left(\gamma_m n^{\nu-\frac{\beta+1}{2}}\|r\|_2\Theta(\sqrt{m})\right) = \Theta\left(\sqrt{m}\gamma_m\|r\|_2 n^{\nu-\frac{\beta}{2}-\frac{1}{2}}\right).$$

$\square$

**Lemma A.3** ($S_2$ Bound). *Let $W$ be the weight matrix (e.g., at step $t$) with unit norm rows. Let $S_2 = \frac{\gamma_m\zeta}{n}qz^T\left[(ra^T)\circ\sigma'_\perp(XW^T)\right]$. Suppose $z, q, \zeta = n^\nu$ are from Assumption 2, $a$ has fixed $\pm 1$ entries (Assumption 3), $\mu = \mathbb{E}_x[\sigma'(Wx)]$ satisfies $\mu_k = \Theta(1)$ (Assumption 4), and the current residual $r$ satisfies $|\frac{z^T r}{n}| = \Theta(\|r\|_2 n^{-\beta/2-1/2})$ (Assumption 6).Then, w.h.p.:*

$$\gamma_m n^{\nu-\frac{\beta}{2}-1}\|r\|_2\sigma_{\min}(\sigma'_\perp(XW^T)) \lesssim \|S_2\|_2 \lesssim \gamma_m\sqrt{m}\|r\|_\infty \min(n^\nu, \|W\|_2 n^{2\nu-\frac{1}{2}}).$$

*Where $\lesssim$ hides universal constants $C_1, C_2$.*

*Proof.* The operator norm is

$$\|S_2\|_2 = \frac{\gamma_m n^\nu}{n}\|z^T((ra^T)\circ\sigma'_\perp(XW^T))\|_2.$$

**Upper Bound:** Using Lemma A.8 and Assumption 3 that $a_i \sim \mathrm{Unif}(\pm 1)$, we have the upper bound

$$\|z^T((ra^T)\circ\sigma'_\perp(XW^T))\|_2 \lesssim \|z\|_2\|r\|_\infty\|a\|_\infty\|\sigma'_\perp(XW^T))\|_2$$
$$\lesssim \|z\|_2\|r\|_\infty\|\sigma'_\perp(XW^T))\|_2.$$

Then with probability $1 - o(1)$, since $z \sim \mathcal{N}(0, I)$, we have that $\|z\|_2 \lesssim C\sqrt{n}$. Hence we get that

$$\|z^T((ra^T)\circ\sigma'_\perp(XW^T))\|_2 \lesssim C\|r\|_\infty\|\sigma'_\perp(XW^T))\|_2\sqrt{n}.$$

Then since we have Assumption 4, we can use Lemma A.7 to bound the norm $\|\sigma'_\perp(XW^T)\|_2$, which gives us that with probability $1 - o(1)$,

$$\|z^T((ra^T)\circ\sigma'_\perp(XW^T))\|_2 \lesssim C\|r\|_\infty\sqrt{n}\min\left(n, \sqrt{n}\|W\Sigma^{1/2}\|_2\right)$$
$$= C\|r\|_\infty\sqrt{n}\min(n, \|W\|_2 n^{\nu+1/2}).$$

Thus, we get that

$$\|S_2\|_2 \lesssim \frac{\gamma_m}{n}n^\nu C\|r\|_\infty\sqrt{n}\min(n, \|W\|_2 n^{\nu+1/2})$$
$$= C\sqrt{m}\gamma_m\|r\|_\infty \min(n^\nu, \|W\|_2 n^{2\nu-\frac{1}{2}}),$$

where we used the proportional scaling of $n$ and $m$, Assumption 1, in the second line.

**Lower Bound:** For a lower bound, we start by writing

$$(ra^T)\circ\sigma'_\perp(XW^T) = \mathrm{Diag}(r)\,\sigma'_\perp(XW^T)\,\mathrm{Diag}(a).$$

Thus, we have that

$$qz^T\left((ra^T)\circ\sigma'_\perp(XW^T)\right) = q\left(z^T\,\mathrm{Diag}(r)\right)\,\sigma'_\perp(XW^T)\,\mathrm{Diag}(a)$$
$$= q(z\circ r)^T\,\sigma'_\perp(XW^T)\,\mathrm{Diag}(a).$$

Taking the norm and recalling that $\zeta = n^\nu$, we get

$$\|\zeta qz^T\left((ra^T)\circ\sigma'_\perp(XW^T)\right)\|_2 = n^\nu\|q\|\left\|(z\circ r)^T\,\sigma'_\perp(XW^T)\,\mathrm{Diag}(a)\right\|.$$

Since the entries of $a$ are $\pm 1$ and $q$ has unit norm, we have that this is the same as

$$n^\nu\|q\|\left\|(z\circ r)^T\,\sigma'_\perp(XW^T)\right\| = n^\nu\left\|(z\circ r)^T\,\sigma'_\perp(XW^T)\right\|.$$

By Cauchy-Schwarz, we have using Assumption 6 $|z^T r/\sqrt{n}\|r\|_2| = \Theta(d^{-\beta/2})$ that

$$\|z\circ r\| = \sqrt{\sum_{i=1}^n (z_i r_i)^2} \geq \frac{|\sum_{i=1}^n z_i r_i|}{\sqrt{\sum_{i=1}^n 1}} = \frac{|z^T r|\|r\|_2}{\sqrt{n}\|r\|_2} = \Omega(n^{-\frac{\beta}{2}}\|r\|_2).$$

Thus, we get that for some constant $C$

$$\|S_2\| \gtrsim C\gamma_m\frac{1}{n}n^{\nu-\frac{\beta}{2}}\|r\|_2\sigma_{\min}(\sigma'_\perp(XW^T)).$$

$\square$

**Lemma A.4** (Upper Bound on $E$). *Assuming Assumption 1], Assumption 3, Assumption 2, and Assumption 4, we have that with probability at least $1 - o(1)$*

$$\|E\|_2 \lesssim C\sqrt{m}\gamma_m\|r\|_\infty\min\left(1, n^{\nu-\frac{1}{2}}\|W\|_2\right).$$

*Proof.* Recall $E = \frac{\gamma_m}{n}X_B^T((ra^T)\circ\sigma'_\perp(XW^T))$. Using Lemma A.8, we have that

$$\frac{n}{\gamma_m}\|E\|_2 \lesssim \|X_B\|_2\|r\|_\infty\|a\|_\infty\|\sigma'_\perp(XW^T)\|_2.$$

Then using Assumption 2, whereby the rows of $X_B$ are iid from $\mathcal{N}(0, \hat{\Sigma})$, we have with probability $1 - o(1)$ that

$$\|X_B\|_2 \lesssim C\sqrt{n},$$

and using Assumption 3, we trivially have that

$$\|a\|_\infty = 1.$$

Thus, we have that

$$\frac{n}{\gamma_m}\|E\|_2 \lesssim C\sqrt{n}\|r\|_\infty\|\sigma'_\perp(XW^T)\|_2.$$

Then using Lemma A.7, we have that with probability $1 - o(1)$

$$\|\sigma'_\perp(XW^T)\|_2 \lesssim C\min\left(n, \sqrt{n}\|W\Sigma^{1/2}\|_2\right).$$

Since $\|\Sigma^{1/2}\| = n^\nu$, we get the result in the proportional scaling of Assumption 1. $\square$

**Theorem 3.1** (Gradient approximation). *Suppose Assumptions 1, 2, 3, 4, 5, 6 are satisfied, $X$ and $W$ are independent, and $\sigma$ is a $\mathcal{C}^2$ function. Define $E = G - S_1 - S_{12} - S_2$. Then, for all $\nu, \alpha \in \mathbb{R}_{\geq 0}$,*

$$\frac{\|G - S_1 - S_{12}\|_2}{\sqrt{m}\gamma_m\|r\|_\infty} = O\left(\|W\|_2 n^{2\nu-\frac{1}{2}}\right), \quad \frac{\|G - S_1 - S_{12} - S_2\|_2}{\sqrt{m}\gamma_m\|r\|_\infty} = O\left(\|W\|_2 n^{\nu-\frac{1}{2}}\right) \quad (4)$$

*with probability $1 - o(1)$ as $d, n, m \to \infty$. Moreover, if $\nu < \frac{1}{2}$ then with the same probability*

$$\frac{\|S_1\|_2}{\|E\|_2} = \Omega\left(\frac{n^{\frac{1}{2}-\nu-\frac{\alpha}{2}}}{\log n\|W\|_2}\right), \quad \frac{\|S_2\|_2}{\|E\|_2} = \Omega\left(\frac{n^\nu}{\log n}\frac{\|(z\circ r)^T\sigma'_\perp(XW^T)\|_2}{\|\sigma'_\perp(XW^T)\|_2}\right), \quad (5)$$

$$\frac{\|S_{12}\|_2}{\|E\|_2} = \Omega\left(\frac{n^{\frac{1}{2}-\frac{\beta}{2}}}{\log n\|W\|_2}\right), \quad \Omega(n^{\nu-\frac{\beta}{2}}) \leq \frac{\|S_{12}\|_2}{\|S_1\|_2} \leq O(n^{\nu-\frac{\beta}{2}+\frac{\alpha}{2}}). \quad (6)$$

*Proof.* We start with the gradient decomposition derived in Section 3:

$$G = S_1 + S_{12} + S_2 + E$$

where

$$S_1 = \gamma_m \frac{X_B^T r}{n} (a \circ \mu)^T$$

$$S_{12} = \gamma_m \zeta \left( \frac{z^T r}{n} \right) q(a \circ \mu)^T$$

$$S_2 = \frac{\gamma_m \zeta}{n} q \left( z^T ((ra^T) \circ \sigma'_\perp (XW^T)) \right)$$

$$E = \frac{\gamma_m}{n} X_B^T ((ra^T) \circ \sigma'_\perp (XW^T)).$$

We assume the conditions of the theorem hold, including the scaling $\sqrt{m}\gamma_m = O(1)$ and the residual concentration $\|r\|_2 / \|r\|_\infty = \Theta(\sqrt{n}/\log n)$ (Assumption 5).

**Proof of Upper Bounds:**

For the first upper bound, we have $G - S_1 - S_{12} - S_2 = E$. Using the upper bound on $\|E\|_2$ from Lemma A.4 and the assumption $\sqrt{m}\gamma_m = O(1)$:

$$\frac{\|G - S_1 - S_{12} - S_2\|_2}{\|r\|_\infty} = \frac{\|E\|_2}{\|r\|_\infty}$$

$$\lesssim \frac{C\sqrt{m}\gamma_m \min \left( 1, n^{\nu - \frac{1}{2}} \|W\|_2 \right)}{\|r\|_\infty}$$

$$= O\left( \min(1, \|W\|_2 n^{\nu - \frac{1}{2}}) \right).$$

For the second upper bound, we have $G - S_1 - S_{12} = S_2 + E$. Using the triangle inequality and the upper bounds on $\|S_2\|_2$ from Lemma A.3 and $\|E\|_2$ from Lemma A.4, along with $\sqrt{m}\gamma_m = O(1)$:

$$\frac{\|G - S_1 - S_{12}\|_2}{\|r\|_\infty} \leq \frac{\|S_2\|_2 + \|E\|_2}{\|r\|_\infty}$$

$$\lesssim \frac{\sqrt{m}\gamma_m \|r\|_\infty \min(n^\nu, \|W\|_2 n^{2\nu - \frac{1}{2}}) + \sqrt{m}\gamma_m \|r\|_\infty \min \left( 1, n^{\nu - \frac{1}{2}} \|W\|_2 \right)}{\|r\|_\infty}$$

$$= O\left( \min(n^\nu, \|W\|_2 n^{2\nu - \frac{1}{2}}) + \min(1, \|W\|_2 n^{\nu - \frac{1}{2}}) \right).$$

**Proof of Lower Bounds:**

We establish lower bounds for the ratios $\|S_1\|/\|E\|$, $\|S_{12}\|/\|E\|$, and $\|S_2\|/\|E\|$. These rely on the lower bounds for $\|S_1\|, \|S_{12}\|, \|S_2\|$ and the upper bound for $\|E\|$. We use the result $\|r\|_2 / \|r\|_\infty = \Theta(\sqrt{n}/\log n)$.

*Ratio $\|S_1\|/\|E\|$:* Using Lemma A.1 (lower bound) and Lemma A.4 (upper bound), we have that

$$\frac{\|S_1\|_2}{\|E\|_2} \gtrsim \frac{\sqrt{m}\gamma_m \mu_{\min} \|r\|_2 n^{-\frac{\alpha+1}{2}}}{\sqrt{m}\gamma_m \|r\|_\infty \min \left( 1, n^{\nu - \frac{1}{2}} \|W\|_2 \right)}$$

$$\gtrsim \frac{\|r\|_2}{\|r\|_\infty} \frac{n^{-(\alpha+1)/2}}{\min(1, \|W\|_2 n^{\nu - 1/2})}$$

$$= \frac{\sqrt{n}}{\log n} \frac{n^{-(\alpha+1)/2}}{\min(1, \|W\|_2 n^{\nu - 1/2})}$$

$$= \frac{n^{-\alpha/2}}{\log n \, \min(1, \|W\|_2 n^{\nu - 1/2})}.$$

If $\nu < 1/2$ and we assume $\|W\|_2 n^{\nu-1/2} = O(1)$ is the dominant term in the minimum, the ratio is

$$\Omega \left( \frac{n^{1/2-\nu-\alpha/2}}{\log n \|W\|_2} \right).$$

If $\nu \geq 1/2$ and assume $\|W\|_2 n^{\nu-1/2} \geq \Omega(1)$, the minimum is $O(1)$. The ratio is

$$\Omega \left( \frac{n^{-\alpha/2}}{\log n} \right).$$

*Ratio $\|S_{12}\|/\|E\|$:* Using Lemma A.2 (lower bound) and Lemma A.4 (upper bound):

$$
\begin{aligned}
\frac{\|S_{12}\|_2}{\|E\|_2} &\gtrsim \frac{\sqrt{m}\gamma_m \|r\|_2 n^{\nu-\beta/2-1/2}}{\sqrt{m}\gamma_m \|r\|_\infty \min(1, \|W\|_2 n^{\nu-1/2})} \\
&\gtrsim \frac{\|r\|_2}{\|r\|_\infty} \frac{n^{\nu-\beta/2-1/2}}{\min(1, \|W\|_2 n^{\nu-1/2})} \\
&= \frac{\sqrt{n}}{\log n} \frac{n^{\nu-\beta/2-1/2}}{\min(1, \|W\|_2 n^{\nu-1/2})} \\
&= \frac{n^{\nu-\beta/2}}{\log n \min(1, \|W\|_2 n^{\nu-1/2})}.
\end{aligned}
$$

If $\nu < 1/2$ and assume $\|W\|_2 n^{\nu-1/2} = O(1)$ dominates the minimum, the ratio is

$$\Omega \left( \frac{n^{1/2-\beta/2}}{\log n \|W\|_2} \right).$$

If $\nu \geq 1/2$ and assume $\|W\|_2 n^{\nu-1/2} \geq \Omega(1)$, the minimum is $O(1)$. The ratio is

$$\Omega \left( \frac{n^{\nu-\beta/2}}{\log n} \right).$$

*Ratio $\|S_2\|/\|E\|$:* We have that

$$
\begin{aligned}
\frac{\|S_2\|}{\|E\|} &\gtrsim \frac{\frac{\gamma_m}{n} n^\nu \|(z \circ r)^T \sigma'_\perp(XW^T)\|}{\frac{\gamma_m}{n} \|X_B\|_2 \|r\|_\infty \|\sigma'_\perp(XW^T)\|} \\
&\gtrsim n^{\nu-\frac{1}{2}} \frac{\|z \circ r\|}{\|r\|_\infty} \kappa\left(\sigma'_\perp(XW^T)\right) \\
&\gtrsim n^{\nu-\frac{1}{2}-\frac{\beta}{2}} \frac{\|r\|_2}{\|r\|_\infty} \kappa\left(\sigma'_\perp(XW^T)\right) \\
&\gtrsim \frac{n^{\nu-\frac{\beta}{2}}}{\log n} \kappa\left(\sigma'_\perp(XW^T)\right)
\end{aligned}
$$

**Relative Sizes** Next, we prove the relative bounds. First, we have that

$$\frac{\|S_{12}\|}{\|S_1\|} = \frac{\|X_S^T r\| \|a \circ \mu\|}{\|X_B^T r\| \|a \circ \mu\|} = \frac{n^{\nu+\frac{1}{2}-\frac{\beta}{2}} \|r\|_2}{\|X_B^T r\|}$$

Then since

$$n^{-\frac{\alpha}{2}+\frac{1}{2}} \|r\|_2 \lesssim \|X_B^T r\|_2 \lesssim \sqrt{n} \|r\|_2,$$

we get that

$$n^{\nu-\frac{\beta}{2}} \lesssim \frac{\|S_{12}\|}{\|S_1\|} \lesssim n^{\nu-\frac{\beta}{2}+\frac{\alpha}{2}}$$

For the second relative bound, we have that

$$\frac{\|S_{12}\|}{\|S_2\|} = \frac{n^{\nu+\frac{1}{2}-\frac{\beta}{2}}\|r\|_2\|a \circ \mu\|}{n^\nu\|(z \circ r)^T\sigma'_\perp(XW^T)\|} = \Theta\left(\frac{n^{1-\frac{\beta}{2}}\|r\|_2}{\|(z \circ r)^T\sigma'_\perp(XW^T)\|}\right)$$

For a lower bound, we get that

$$\frac{\|S_{12}\|}{\|S_2\|} \gtrsim C\frac{\|r\|_2}{\|z\|_2\|r\|_2}\frac{n^{1-\frac{\beta}{2}}}{n^{\nu+\frac{1}{2}}} = \frac{1}{n^{\nu+\frac{\beta}{2}}}$$

For an upper bound, we have that

$$\frac{\|S_{12}\|}{\|S_2\|} \lesssim \frac{n^{1-\frac{\beta}{2}}\|r\|_2}{n^{-\frac{\beta}{2}}\|r\|_2\sigma_{\min}(\sigma'_\perp(XW^T))} = \frac{n}{\sigma_{\min}(\sigma'_\perp(XW^T))}$$

$\square$

**Theorem 3.2** (Large data-spike gradient approximation). *Suppose Assumptions 1, 2, 3, 4, 5, and 6 are satisfied, and define $E_L = G - S_{12} - S_2$. Then, with probability $1 - o(1)$ for $\nu \geq \frac{1}{2}$ we have*

$$\frac{\|E_L\|_2}{\sqrt{m}\gamma_m\|r\|_\infty} = O(1), \frac{\|S_{12}\|_2}{\|E_L\|_2} = \Omega\left(\frac{n^{\nu-\frac{\beta}{2}}}{\log n}\right), \frac{\|S_2\|_2}{\|E_L\|_2} = \Omega\left(\frac{n^\nu}{\log n}\frac{\|(z \circ r)^T\sigma'_\perp(XW^T)\|_2}{\|\sigma'_\perp(XW^T)\|_2}\right).$$

(8)

*Proof.* This proof is exactly the same as Theorem 3.1. In particular, we note that

$$E_L = E + S_1$$

Except we use the following upper bounds. We have already bounded $S_1$, in the following we bound $E$.

**Data Spike:** The operator norm is

$$\|S_2\|_2 = \frac{\gamma_m n^\nu}{n}\|z^T((ra^T) \circ \sigma'_\perp(XW^T))\|_2.$$

Using Lemma A.8 and Assumption 3 that $a_i \sim \text{Unif}(\pm 1)$, we have the upper bound

$$\|z^T((ra^T) \circ \sigma'_\perp(XW^T))\|_2 \lesssim \|z\|_2\|r\|_\infty\|a\|_\infty\|\sigma'_\perp(XW^T))\|_2$$
$$\lesssim \|z\|_2\|r\|_\infty\|\sigma'_\perp(XW^T))\|_2.$$

Then with probability $1 - o(1)$, since $z \sim \mathcal{N}(0, I)$, we have that $\|z\|_2 \lesssim C\sqrt{n}$. Hence we get that

$$\|z^T((ra^T) \circ \sigma'_\perp(XW^T))\|_2 \lesssim C\|r\|_\infty\|\sigma'_\perp(XW^T))\|_2\sqrt{n}.$$

Then since we have Assumption 4, we can bound the norm $\|\sigma'_\perp(XW^T)\|_2$ by $O(n)$

$$\|z^T((ra^T) \circ \sigma'_\perp(XW^T))\|_2 \lesssim C\|r\|_\infty\sqrt{n}n$$

Thus, we get that

$$\|S_2\|_2 \lesssim C\sqrt{m}\gamma_m\|r\|_\infty n^\nu,$$

where we used the proportional scaling of $n$ and $m$, Assumption 1, in the second line.

**Error Term:** Recall $E = \frac{\gamma_m}{n}X_B^T((ra^T) \circ \sigma'_\perp(XW^T))$. Using Lemma A.8, we have that

$$\frac{n}{\gamma_m}\|E\|_2 \lesssim \|X_B\|_2\|r\|_\infty\|a\|_\infty\|\sigma'_\perp(XW^T)\|_2.$$

Then using Assumption 2, whereby the rows of $X_B$ are iid from $\mathcal{N}(0, \hat{\Sigma})$, we have with probability $1 - o(1)$ that

$$\|X_B\|_2 \lesssim C\sqrt{n},$$

and using Assumption 3, we trivially have that

$$\|a\|_\infty = 1.$$

Thus, we have that

$$\frac{n}{\gamma_m}\|E\|_2 \lesssim C\sqrt{n}\|r\|_\infty\|\sigma'_\perp(XW^T)\|_2.$$

Then

$$\|\sigma'_\perp(XW^T)\|_2 \leq O(n).$$

Since $\|\Sigma^{1/2}\| = n^\nu$, we get the result in the proportional scaling of Assumption 1. $\square$

### A.2.2 Helper Results: Subgaussianity and Concentration

**Lemma A.5.** *Let $Z \in \mathbb{R}^{n \times d}$ be a matrix with standard normal IID entries. If $n < d$, then as $n/d \to c \in (0, 1)$, we have that with probability 1, the eigenvalues of $\frac{1}{d}ZZ^T$ are $\Theta(1)$. Further,*

$$\sigma_{\min}(Z) = \Theta(\sqrt{d} - \sqrt{n}), \quad \sigma_{\max}(Z) = \Theta(\sqrt{d} + \sqrt{n}).$$

*Proof.* As $\frac{1}{d}ZZ^T$ is a Wishart matrix, the limiting empirical spectral distribution almost surely weakly converges to the Marchenko-Pastur distribution supported on $[(1 - \sqrt{c})^2, (1 + \sqrt{c})^2]$. $\square$

**Lemma A.6.** *Let $X_B \in \mathbb{R}^{n \times d}$ have IID rows from $\mathcal{N}(0, \hat{\Sigma})$, where $\lambda_k(\hat{\Sigma}) \sim k^{-\alpha}$ as per Assumption 2. Then with probability $1 - 2\exp(-cn)$ for positive universal constants c, we have that*

$$\Omega\left(n^{\frac{1-\alpha}{2}}\right) \leq \|X_B\|_2 \leq O\left(n^{\frac{1}{2}}\right)$$

*Proof.* We can write $X_B = \hat{\Sigma}^{1/2}Z$ where $Z \in \mathbb{R}^{n \times d}$ has IID standard normal entries. Using Lemma A.5, we have that in the proportional regime (Assumption 1), $\|Z\|_2 = \Theta(\sqrt{n})$. The result follows using the fact that

$$\sigma_{\min}(\hat{\Sigma}^{1/2})\|Z\|_2 \leq \|X_B\|_2 = \|\hat{\Sigma}^{1/2}Z\|_2 \leq \sigma_{\max}(\hat{\Sigma}^{1/2})\|Z\|_2,$$

and noting that

$$\sigma_{\min}(\Sigma^{1/2}) = \Theta(n^{-\alpha/2}) \text{ and } \sigma_{\max}(\Sigma^{1/2}) = \Theta(1).$$

$\square$

**Lemma A.7.** *Let $W$ be a given fixed matrix indepedent of $X$. If Assumption 4 is satisfied and $\sigma$ is $\mathcal{C}^2$, then we have with probability $1 - C\exp(-cn)$ for positive universal constants $c, C$, that*

$$\|\sigma'_\perp(XW^T)\|_2 \lesssim C' \min\left(n, \sqrt{n}\|W\Sigma^{1/2}\|_2\right).$$

*for some constant $C' > 0$. Here $\Sigma = \hat{\Sigma} + \zeta^2 qq^T$ is the full data covariance from Assumption 2.*

*Proof.* Since $\sigma$ is $L$-Lipschitz (Assumption 4), its derivative $\sigma'$ is bounded by $L$. As $\mu = \mathbb{E}_x[\sigma'(Wx)]$, the centered term $\sigma'_\perp(XW^T) = \sigma'(XW^T) - \mathbf{1}_n\mu^T$ has entries bounded by some $M$ (e.g., $M = 2L$). Thus, using the relation between operator and Frobenius norms:

$$\|\sigma'_\perp(XW^T)\|_2^2 \leq \|\sigma'_\perp(XW^T)\|_F^2 \leq Mnm.$$

Thus, we have that in the proportional regime

$$\|\sigma'_\perp(XW^T)\|_2 = O(n).$$

On the other hand, $\sigma'_\perp(XW^T)$ represents mean-centered features and is Lipschitz, using Corollary A.2, with probability $1 - C\exp(-cn)$, we have that

$$\|\sigma'_\perp(XW^T)\|_2 = O\left(\sqrt{n}\|W\Sigma^{1/2}\|_2\right).$$

The overall bound follows by taking the minimum of the two derived bounds. $\square$

**Lemma A.8.** *For any vectors $u, v$ and matrix $A$, we have that*

$$\min_i |u_i| \min_j |v_j| \|A\|_2 \leq \|(uv^T) \circ A\|_2 \leq \|u\|_\infty \|v\|_\infty \|A\|_2.$$

*Proof.* This follows from the observation that

$$(uv^T) \circ A = \text{diag}(u)A\text{diag}(v),$$

where $\text{diag}(u)$ is the diagonal matrix with $u$ in the diagonal. Then using the fact that

$$\sigma_{\min}(B)\|A\|_2 \leq \|AB\|_2 \leq \sigma_{\max}(B)\|A\|_2,$$

where $\sigma_{\min}$ is allowed to be zero and noticing that

$$\sigma_{\max}(\text{diag}(u)) = \|u\|_\infty \quad \text{and} \quad \sigma_{\min}(\text{diag}(u)) = \min_i |u_i|.$$

The bounds follow from applying the matrix norm inequality twice. $\square$

**Lemma A.9** (Sub-Gaussianity). *For $x \sim \mathcal{N}(0, \Sigma)$, a fixed vector $w \in \mathbb{R}^d$, and an $\mathcal{L}_f$-Lipschitz function $f : \mathbb{R} \to \mathbb{R}$, the random variable $f(w^T x)$ is sub-gaussian with subgaussian norm at most $C\mathcal{L}_f^2 \|w^T \Sigma^{1/2}\|_2^2$ for some constant $C$. Furthermore,*

$$\mathbb{E}[|f(w^T x)|] = |f(0)| + O\left(\mathcal{L}_f \|w^T \Sigma^{1/2}\|_2\right) = O(1 + \mathcal{L}_f \|w^T \Sigma^{1/2}\|_2).$$

*Proof.* Using Lipschitzness,

$$\left| f(x^T w) - f(0^T w) \right| \le \mathcal{L}_f |x^T w - 0| = \mathcal{L}_f |x^T w|.$$

The variable $w^T x \sim \mathcal{N}(0, \sigma_w^2)$ where $\sigma_w^2 = \|w^T \Sigma^{1/2}\|_2^2$. Thus, $w^T x$ is $(\sigma_w^2)$-sub-gaussian. For $t \ge 0$,

$$\Pr\left[|f(x^T w) - f(0)| \ge t\right] \le \Pr\left[\mathcal{L}_f |x^T w| \ge t\right] \le 2 \exp\left(-\frac{t^2}{2\mathcal{L}_f^2 \|w^T \Sigma^{1/2}\|_2^2}\right).$$

Thus, we see that

$$\Pr[|f(x^T w)| \ge t] \le 2 \exp\left(-\frac{(t-c)^2}{2\mathcal{L}_f^2 \|w^T \Sigma^{1/2}\|_2^2}\right),$$

where $c = |f(0)|$. For the expectations, taking expectations, we get that

$$\mathbb{E}\left[|f(x^T w) - f(0)|\right] \le \mathbb{E}\left[\mathcal{L}_f |x^T w|\right] = \mathcal{L}_f \sqrt{\frac{2}{\pi}} \|w^T \Sigma^{1/2}\|_2^2.$$

Using $|f(w^T x)| \le |f(w^T x) - f(0)| + |f(0)|$ and the triangle inequality for expectations, $\mathbb{E}[|f(w^T x)|] \le \mathbb{E}[|f(w^T x) - f(0)|] + |f(0)| = |f(0)| + O(\mathcal{L}_f \sigma_w)$, giving the result. $\square$

**Lemma A.10** (Covariance Operator Norm Bound). *Let $W \in \mathbb{R}^{m \times d}$ be a fixed matrix whose rows have unit norm and let $x \sim \mathcal{N}(0, \Sigma)$. Suppose that $f : \mathbb{R} \to \mathbb{R}$ is $\mathcal{L}_f$ Lipschitz respectively. Define the population second moment matrix*

$$\Phi = \mathbb{E}_x[f(Wx)f(Wx)^T],$$

*where $f$ is applied element-wise to the vector $Wx \in \mathbb{R}^m$. Then*

$$\|\Phi\|_2 \le \|\mathbb{E}_x[f(Wx)]\|_2^2 + \|W\Sigma^{1/2}\|_2^2 \mathcal{L}_f^2$$

*for some universal constants $C_1, C_2$.*

*Proof.* We note that $\Phi$ is the uncentered covariance matrix. However, to bound the operator norm of $\Phi$ we need to consider the centered covariance matrix $\check{\Phi}$

$$\check{\Phi} = \underbrace{\mathbb{E}\left[f(Wx)f(Wx)^T\right]}_{\Phi} - \mathbb{E}[f(Wx)]\,\mathbb{E}[f(Wx)]^T$$

Then we see that

$$
\begin{aligned}
\left\|\check{\Phi}\right\|_2 &= \sup_{\|v\|=1} v^T \check{\Phi} v \\
&= \sup_{\|v\|=1} v^T \Phi v - \left(\mathbb{E}\left[v^T f(Wx)\right]\right)^2 \\
&= \sup_{\|v\|=1} \mathbb{E}\left[\left(v^T f(Wx)\right)\left(v^T f(Wx)\right)^T\right] - \left(\mathbb{E}\left[v^T f(Wx)\right]\right)^2 \\
&= \sup_{\|v\|=1} \mathrm{Var}\left(v^T f(Wx)\right)
\end{aligned}
$$

We want to bound this using the Gaussian Poincare inequality. Which we recall here (Link). Let $g : \mathbb{R}^d \to \mathbb{R}$ be a $C^1$ function then

$$\mathrm{Var}_{z \sim \mathcal{N}(0,I)}(g(z)) \le \mathbb{E}_{z \sim \mathcal{N}(0,I)}\left[\|\nabla g(z)\|^2\right]$$

Since $x \sim \mathcal{N}(0, \Sigma)$, we can write it as $x = \Sigma^{1/2} z$. Thus, define the function

$$g(z) := f(Wx) = v^T f\left(W\Sigma^{1/2}x\right) = \sum_{k=1}^{m} v_k f\left(w_k^T \Sigma^{1/2}x\right).$$

Let us then define

$$u = \left[v_1 f'\left(w_1^T \Sigma^{1/2}x\right) \quad \ldots \quad v_m f'\left(w_m^T \Sigma^{1/2}x\right)\right]^T$$

Then we see that

$$\nabla g(z)^T = \sum_{k=1}^{m} v_k f'\left(w_k^T \Sigma^{1/2}x\right)\left(w_k^T \Sigma^{1/2}\right) = u^T W \Sigma^{1/2}$$

Thus, we see that

$$\mathbb{E}_z\left[\|\nabla_z g(z)\|^2\right] \leq \mathbb{E}_x\left[\|W\Sigma^{1/2}\|_2^2\|u\|^2\right] \leq \|W\Sigma^{1/2}\|_2^2 \mathbb{E}_x\left[\|u\|^2\right]$$

Then using Lemma A.9 and noting that $f'$ is bounded by $\mathcal{L}_f$, we get that

$$\mathbb{E}_x\left[\sum_{k=1}^{m} u_k^2\right] = \sum_{k=1}^{m} v_k^2 \mathbb{E}_x\left[\left(f'(w_k^T x)\right)^2\right]$$

$$\leq \sum_{k=1}^{m} v_k^2 \mathcal{L}_f^2 \qquad\qquad \leq \mathcal{L}_f^2$$

Thus, we have that

$$\mathbb{E}\left[\|\nabla g(z)\|^2\right] \leq \|W\Sigma^{1/2}\|_2^2 \mathcal{L}_f^2$$

Thus, using the Gaussian Poincare inequality, we see that

$$\left\|\check{\Phi}\right\|_2 \leq \|W\Sigma^{1/2}\|_2^2 \mathcal{L}_f^2$$

Thus, we see that

$$\|\Phi\|_2 \leq \|\check{\Phi} - \Phi\|_2 + \|W\Sigma^{1/2}\|_2^2 \mathcal{L}_f^2$$

Finally, we see that

$$\left\|\check{\Phi} - \Phi\right\|_2 = \left\|\mathbb{E}\left[f(Wx)\right]\mathbb{E}\left[f(Wx)\right]^T\right\|_2$$

$$= \|\mathbb{E}\left[f(Wx)\right]\|_2^2$$

Thus,

$$\|\Phi\|_2 \leq \|\mathbb{E}\left[f(Wx)\right]\|_2^2 + \|W\Sigma^{1/2}\|_2^2 \cdot \mathcal{L}_f^2$$

$\qquad\qquad\qquad\qquad\qquad\qquad\qquad\qquad\qquad\qquad\qquad\qquad\qquad\qquad\qquad$ $\square$

We are going to instantiate a few corollaries for cases that we care about. Specifically, we shall $f = \sigma'_\perp$ as the non-linearity. In this case we have that $\mathbb{E}\left[f(Wx)\right] = 0$.

**Corollary A.1.** *If $\mathbb{E}\left[f(Wx)\right] = 0$, we have that*

$$\|\Phi\|_2 \leq \|W\Sigma^{1/2}\|_2^2 \mathcal{L}_f^2.$$

We shall also need to bound the norm of the expectation. In the case, when $\sigma$ is bounded, we get that the expectation

**Lemma A.11** (Feature Norm Bound). *Let $x_i \sim \mathcal{N}(0, \Sigma)$ be IID for $i = 1 \ldots n$, forming rows of $X$. Let $W \in \mathbb{R}^{m \times d}$ be a fixed matrix whose rows $w_j$ have norm $\|w_j\|_2 = 1$. Let $f : \mathbb{R} \to \mathbb{R}$ be $\mathcal{L}_f$-Lipschitz. Define the population second moment matrix*

$$\Phi = \mathbb{E}_x[f(Wx)f(Wx)^T]$$

*(as in Lemma A.10). Then with probability $1 - 2e^{-cn}$ for some universal constant $c > 0$,*

$$\left\|\frac{1}{\sqrt{n}}f(XW^T)\right\|_2 \leq \left(1 + C'\sqrt{\frac{m}{n}}\right)\sqrt{\|\Phi\|_2}$$

*for some universal constant $C'$.*

*Proof.* Since $x_i$ are IID, we have the rows of $f(XW^T) \in \mathbb{R}^{n \times m}$ are IID. Additionally, by Lemma A.9 the entries are $\mathcal{L}_f^2 \|w_i^T \Sigma^{1/2}\|_2^2$ sub-gaussian entries. Thus, we have that

$$\check{X} = \frac{1}{\mathcal{L}_f \max_{i=1...m} \|w_i^T \Sigma^{1/2}\|_2} f(XW^T)$$

has IID rows whose sub-Gaussian norm is at most a universal constant. Let

$$\check{\Phi} = \frac{1}{n} \mathbb{E}\left[\check{X}^T \check{X}\right] = \frac{1}{\mathcal{L}_f^2 \max_{i=1...m} \|w_i^T \Sigma^{1/2}\|_2^2} \Phi$$

Then using Equation 5.26 from [42], there exists universal constant $C, c$ such that

$$\Pr\left[\left\|\frac{1}{n}\check{X}^T \check{X} - \check{\Phi}\right\|_2 \geq \max(\delta, \delta^2)\|\check{\Phi}\|_2\right] < 2e^{-ct^2}, \quad \delta = C\sqrt{\frac{m}{n}} + \frac{t}{\sqrt{n}}$$

Thus, with probability $1 - 2e^{-ct^2}$, we have that

$$\left\|\frac{1}{n}\check{X}^T \check{X} - \check{\Phi}\right\|_2 \leq \max(\delta, \delta^2)\|\check{\Phi}\|_2$$

Using the reverse triangle inequality, we have that

$$\frac{1}{n}\|\check{X}^T \check{X}\|_2 \leq \left\|\frac{1}{n}\check{X}^T \check{X} - \check{\Phi}\right\|_2 + \|\check{\Phi}\|_2$$

Thus, with probability at least $1 - 2e^{-ct^2}$, we have that

$$\frac{1}{n}\|\check{X}^T \check{X}\|_2 \leq \|\check{\Phi}\|_2 + \max(\delta, \delta^2)\|\check{\Phi}\|_2$$

Thus, we get that

$$\frac{1}{\sqrt{n}}\|\check{X}\|_2 \leq \sqrt{\|\check{\Phi}\|_2 + \max(\delta, \delta^2)\|\check{\Phi}\|_2}$$

Multiplying both sides by $\mathcal{L}_f \max_{i=1...m} \|w_i^T \Sigma^{1/2}\|_2$, we see that

$$\left\|\frac{1}{\sqrt{n}}f(W_0\tilde{X}^T)\right\|_2 \leq \mathcal{L}_f \max_{i=1...m} \|w_i^T \Sigma^{1/2}\|_2(1 + C'\delta)\sqrt{\|\check{\Phi}\|_2}$$

$$\leq (1 + C'\delta)\sqrt{\mathcal{L}_f^2 \max_{i=1...m} \|w_i^T \Sigma^{1/2}\|_2^2\|\check{\Phi}\|_2}$$

$$\leq (1 + C'\delta)\sqrt{\|\Phi\|_2}$$

Using $t = \sqrt{m}$, we see that with probability $1 - 2e^{-cm}$,

$$\left\|\frac{1}{\sqrt{n}}f(W_0\tilde{X}^T)\right\|_2 \leq \left(1 + C'\sqrt{\frac{m}{n}}\right)\sqrt{\|\Phi\|_2}$$

$\square$

Hence, we can again instantiate some simple corollaries.

**Corollary A.2.** *If $\mathbb{E}[f(Wx)] = 0$, we have that*

$$\left\|f(W_0\tilde{X}^T)\right\|_2 \leq \mathcal{L}_f C\|W\Sigma^{1/2}\|_2\sqrt{n}$$

Another important case, if $f$ is uniformly bounded. This is the case, when we apply it for $\sigma', \sigma''$. Here we either have the expectation is zero. In which Corollary A.2 applies. If the mean in non-zero then we get the following.

**Corollary A.3.** *If $|\mathbb{E}[f(z)]| = M$, we have that*

$$\left\|f(W_0\tilde{X}^T)\right\|_2 \leq C\left[n + \mathcal{L}_f\|W\Sigma^{1/2}\|_2\sqrt{n}\right].$$

## A.3 ReLU Data Alignment

**Lemma A.12.** *Let $M = uv^T$ be a non-zero rank 1 matrix, where $u \in \mathbb{R}^m$ and $v \in \mathbb{R}^n$. Assume that all entries of $u$ and $v$ are non-zero, i.e., $u_i \neq 0$ for all $i = 1, \ldots, m$ and $v_j \neq 0$ for all $j = 1, \ldots, n$. Let $\tilde{M}$ be the matrix with entries $\tilde{M}_{ij} = \delta_{u_i v_j > 0}$. Let $\hat{M} = \tilde{M} - 0.5J$, where $J$ is the $m \times n$ matrix of all ones. Then, $\mathrm{rank}(\hat{M}) = 1$.*

*Proof.* Let $u', u'' \in \{0, 1\}^m$ and $v', v'' \in \{0, 1\}^n$ be indicator vectors defined as follows:

- $u'_i = \delta_{u_i > 0}$

- $u''_i = \delta_{u_i < 0}$

- $v'_j = \delta_{v_j > 0}$

- $v''_j = \delta_{v_j < 0}$

Since we assume $u_i \neq 0$ and $v_j \neq 0$ for all $i, j$, every entry in $u$ is either positive or negative, and similarly for $v$. This means $1_m = u' + u''$ and $1_n = v' + v''$, where $1$ denotes a vector of all ones of the appropriate dimension.

The entry $\tilde{M}_{ij} = \delta_{u_i v_j > 0}$ is 1 if and only if ($u_i > 0$ and $v_j > 0$) or ($u_i < 0$ and $v_j < 0$). This can be written as:

$$\tilde{M} = u'(v')^T + u''(v'')^T$$

The all-ones matrix $J$ can be written as $J = 1_m 1_n^T$. Using the property that $1 = u' + u''$ and $1 = v' + v''$:

$$J = (u' + u'')(v' + v'')^T$$
$$= u'(v')^T + u'(v'')^T + u''(v')^T + u''(v'')^T$$

Now we compute $\hat{M} = \tilde{M} - 0.5J$:

$$\hat{M} = (u'(v')^T + u''(v'')^T) - 0.5(u'(v')^T + u'(v'')^T + u''(v')^T + u''(v'')^T)$$
$$= 0.5u'(v')^T + 0.5u''(v'')^T - 0.5u'(v'')^T - 0.5u''(v')^T$$
$$= 0.5\left[u'(v')^T - u'(v'')^T - u''(v')^T + u''(v'')^T\right]$$
$$= 0.5\left[u'((v')^T - (v'')^T) - u''((v')^T - (v'')^T)\right]$$
$$= 0.5(u' - u'')((v')^T - (v'')^T)$$
$$= 0.5(u' - u'')(v' - v'')^T$$

Let $\mathrm{sign}(u)$ denote the vector with entries $\mathrm{sign}(u_i)$, where $\mathrm{sign}(x) = 1$ if $x > 0$ and $\mathrm{sign}(x) = -1$ if $x < 0$. Since no $u_i$ is zero, $(u' - u'')_i = \delta_{u_i > 0} - \delta_{u_i < 0} = \mathrm{sign}(u_i)$. Similarly, $(v' - v'')_j = \mathrm{sign}(v_j)$. Thus, we have shown:

$$\hat{M} = 0.5 \cdot \mathrm{sign}(u) \cdot \mathrm{sign}(v)^T$$

Since $M = uv^T$ is non-zero, both $u$ and $v$ must be non-zero vectors. Because we assumed no zero entries, the vectors $\mathrm{sign}(u)$ (containing only $\pm 1$) and $\mathrm{sign}(v)$ (containing only $\pm 1$) are non-zero vectors. The matrix $\hat{M}$ is expressed as the outer product of two non-zero vectors. Therefore, $\mathrm{rank}(\hat{M}) = 1$. $\qquad\square$

**Proposition 3.1** (ReLU gradient). *If $2\nu > 1 - \alpha$, and the row of $W$ are i.i.d. from the unit sphere, then with probability $1 - o(1)$ we have that $\sigma'_\perp(XW^T) = \frac{1}{2}\mathrm{sign}(z_i)\,\mathrm{sign}(Wq)^T$.*

*Proof.* Recall the data decomposition $x_i = \zeta z_i q + x_{b,i}$, where the *spike direction* $q \in \mathbb{R}^d$ is unit-norm, $z_i \sim \mathcal{N}(0, 1)$, the bulk component $x_{b,i}$ has spectrum exponent $\alpha$, and the spike magnitude scales

as $\zeta = n^\nu$. Since each row $w_k^T$ of $W$ is uniform on $\mathbb{S}^{d-1}$, $\|Wq\|_2 \approx \sqrt{m/d}$ with high probability. Using standard concentration for random projections, with probability $1 - o(1)$,

$$\|Wx_{b,i}\|_2^2 \ \leq \ C \|x_{b,i}\|_2^2 \ = \ C \sum_{j=1}^{d} j^{-\alpha} \ = \ \begin{cases} \Theta(d^{1-\alpha}) & \alpha < 1, \\ \Theta(\log d) & \alpha = 1, \\ O(1) & \alpha > 1. \end{cases} \qquad (14)$$

For the spike term $\| W(\zeta z_i q)\|_2 \ = \ |z_i|\,\zeta\,\|Wq\|_2 \ \gtrsim \ n^\nu \sqrt{\frac{m}{d}}\,|z_i| \ \geq \ n^\nu$, since $|z_i| \geq c$ with probability $1 - o(1)$ for some universal $c > 0$. Hence, whenever $2\nu > 1 - \alpha$, the spike contribution $W(\zeta z_i q)$ dominates the bulk, so that $\mathrm{sign}(Wx_i) = \mathrm{sign}(W(\zeta z_i q))$. Then Lemma A.12 then implies for ReLU that

$$\sigma_\perp'(XW^T) = \tfrac{1}{2}\,\mathrm{sign}(z_i)\,\mathrm{sign}(Wq)^T.$$

$\square$

# B  Assumption Discussion

## B.1  Activation Function Properties

We verify the smoothness and lipschitzness conditions for several common activation functions.

### B.1.1  Sigmoid Function

Let $\sigma(u) = (1 + e^{-u})^{-1}$.

**Smoothness:** The Sigmoid function is infinitely differentiable ($C^\infty$) for all $u \in \mathbb{R}$.

$$\sigma'(u) = \sigma(u)(1 - \sigma(u))$$
$$\sigma''(u) = \sigma'(u)(1 - 2\sigma(u)) = \sigma(u)(1 - \sigma(u))(1 - 2\sigma(u))$$

Both $\sigma'(u)$ and $\sigma''(u)$ exist for all $u \in \mathbb{R}$.

**Lipschitzness:** Since Sigmoid is bounded and all derivatives of the sigmoid can be written as a polynomial of sigmoid, we see that the derivatives are bounded and hence lipschitz.

**Non-Vanishing Derivative** Here we show that if the weight vector $w_j$ is drawn uniformly from the unit sphere $\mathbb{S}^{d-1}$, then the expected derivative $\mu_j = \mathbb{E}_x[\sigma'(w_j^T x)]$ is $\Omega(1)$ when $\nu < 1/2$.

The derivative $\sigma'(u) = \sigma(u)(1 - \sigma(u))$ is bounded. We can see that the argument $u_j = w_j^T x$ is Gaussian $N(0, \sigma_{u_j}^2)$, with variance $\sigma_{u_j}^2 = w_j^T \hat{\Sigma} w_j + n^{2\nu}(w_j^T q)^2$. Then the behavior of $\mu_j$ is such that if $\sigma_{u_j}^2 = O(1)$, then $\mu_j = \Omega(1)$. Specifically, if $\sigma_{u_j}^2 \to 0$, then $\mu_j \to \sigma'(0) = 0.25$. If $\sigma_{u_j}^2 \to \infty$, then $\mu_j \to 0$.

*Spike Contribution* $V_S = n^{2\nu}(w_j^T q)^2$: For a fixed $q \in \mathbb{S}^{d-1}$ and random $w_j \in \mathbb{S}^{d-1}$, the term $(w_j^T q)^2$ concentrates around its mean $\mathbb{E}[(w_j^T q)^2] = 1/d$. With high probability for large $d$, $(w_j^T q)^2 = \Theta(1/d)$. Then in proportional regime, we have that, $V_S = n^{2\nu} \cdot \Theta(1/n) = \Theta(n^{2\nu-1})$. Since $\nu < 1/2$, $2\nu - 1 < 0$, so $V_S = o(1)$ as $n \to \infty$.

*Bulk Contribution* $V_B = w_j^T \hat{\Sigma} w_j$: For random $w_j \in \mathbb{S}^{d-1}$, $w_j^T \hat{\Sigma} w_j$ concentrates around $\mathbb{E}[w_j^T \hat{\Sigma} w_j] = \frac{1}{d}\mathrm{Tr}(\hat{\Sigma})$. The eigenvalues $\lambda_k(\hat{\Sigma}) \sim k^{-\alpha}$.

- If $\alpha = 0$: $\mathrm{Tr}(\hat{\Sigma}) = \Theta(d)$, so $V_B = \Theta(1)$.

- If $0 < \alpha < 1$: $\mathrm{Tr}(\hat{\Sigma}) = \Theta(d^{1-\alpha})$, so $V_B = \Theta(d^{-\alpha}) = \Theta(n^{-\alpha}) = o(1)$.

- If $\alpha = 1$: $\mathrm{Tr}(\hat{\Sigma}) = \Theta(\log d)$, so $V_B = \Theta((\log d)/d) = \Theta((\log n)/n) = o(1)$.

- If $\alpha > 1$: $\mathrm{Tr}(\hat{\Sigma}) = \Theta(1)$, so $V_B = \Theta(1/d) = \Theta(1/n) = o(1)$.

Thus, $V_B$ is either $\Theta(1)$ (for $\alpha = 0$) or $o(1)$ (for $\alpha > 0$).

### B.1.2 Hyperbolic Tangent (Tanh) Function

Let $\sigma(u) = \tanh(u)$.

**Smoothness:** The Tanh function is $C^\infty$ for all $u \in \mathbb{R}$.

$$\sigma'(u) = 1 - \tanh^2(u) = \text{sech}^2(u)$$
$$\sigma''(u) = -2\tanh(u)\text{sech}^2(u)$$

Both $\sigma'(u)$ and $\sigma''(u)$ exist for all $u \in \mathbb{R}$.

**Lipschitzness:**

- For $\sigma(u)$: $\max |\sigma'(u)| = \sigma'(0) = 1$. Thus, $\sigma(u)$ is 1-Lipschitz.

- For $\sigma'(u)$: $\max |\sigma''(u)|$ occurs at $u = \text{arctanh}(\pm 1/\sqrt{3})$, giving $|\sigma''(u)| = \frac{4}{3\sqrt{3}} \approx 0.7698$. Thus, $\sigma'(u)$ is Lipschitz with $L \approx 0.77$ (or $L = 1$ as a looser bound).

$L = 2$ serves as a common upper bound.

**Non-vanishing Derivative:** Let $\sigma(u) = \tanh(u)$. Its derivative is $\sigma'(u) = \text{sech}^2(u)$. This derivative is always positive, $0 < \sigma'(u) \leq 1$, with a maximum of $\sigma'(0) = 1$, and $\sigma'(u) \to 0$ as $|u| \to \infty$. The analysis of the expected derivative $\mu_j = \mathbb{E}_x[\sigma'(w_j^T x)]$ parallels that of the Sigmoid function.

### B.1.3 Rectified Linear Unit (ReLU) Function

Let $\sigma(u) = \max(0, u)$.

**Smoothness:** Here we see that the derivatives for $u \neq 0$ are as follows

$$\sigma'(u) = \begin{cases} 0 & \text{if } u < 0 \\ 1 & \text{if } u > 0 \end{cases}, \qquad \sigma''(u) = 0 \quad \text{for } u \neq 0$$

**Lipschitzness:**

- For $\sigma(u)$: $|\sigma'(u)| \leq 1$ a.e. Thus, $\sigma(u)$ is 1-Lipschitz.
- For $\sigma'(u)$: $\sigma'(u)$ is a step function. It is bounded, but not Lipschitz over $\mathbb{R}$ due to the discontinuity at $u = 0$. However, its values are 0 or 1.

**Non-vanishing Derivative:** Since $Wx$ is symmetric, we get that the mean is 0.5.

### B.1.4 Exponential Linear Unit (ELU) Function

Let $\sigma(u) = \begin{cases} u & \text{if } u > 0 \\ e^u - 1 & \text{if } u \leq 0 \end{cases}$.

**Smoothness:** The derivatives are as follows.

$$\sigma'(u) = \begin{cases} 1 & \text{if } u > 0 \\ e^u & \text{if } u \leq 0 \end{cases} \qquad \sigma''(u) = \begin{cases} 0 & \text{if } u > 0 \\ e^u & \text{if } u < 0 \end{cases}$$

Here we have that $\sigma'$ is continuous, and $\sigma''$ is defined everywhere except for 0.

**Lipschitzness:**

- For $\sigma(u)$: For $u > 0$, $\sigma'(u) = 1$. For $u \leq 0$, $\sigma'(u) = e^u \in (0, 1]$. Thus $|\sigma'(u)| \leq 1$. So $\sigma(u)$ is 1-Lipschitz.

- For $\sigma'(u)$: For $u > 0$, $\sigma''(u) = 0$. For $u < 0$, $\sigma''(u) = e^u \in (0, 1)$. On $[-1, 1]$, the function is continuous. Hence lipschitz. Thus, we have global lipschitzness.

**Non-vanishing Derivative:** The derivative dominates the ReLU case. Hence $\mu_j$ is at least 0.5.

### B.1.5 Swish Function

Let $\sigma(u) = u \cdot \text{sigmoid}(u) = u(1 + e^{-u})^{-1}$.

**Smoothness:** This follows from smoothness of Sigmoid.

**Lipschitzness:** Let $S(u) = \text{sigmoid}(u) = (1 + e^{-u})^{-1}$. Then $\sigma(u) = uS(u)$.

- For $\sigma(u)$: The first derivative is:
$$\sigma'(u) = S(u) + uS'(u) = S(u) + uS(u)(1 - S(u))$$
This is a continuous function that decays to zero. Hence is bounded.

- For $\sigma'(u)$: The second derivative of $\sigma(u)$ is:
$$\sigma''(u) = \frac{d}{du}(S(u) + uS'(u)) = S'(u) + (S'(u) + uS''(u))$$
$$= 2S'(u) + uS''(u)$$
This is a continuous function that decays to zero. Hence is bounded.

- For $\sigma''(u)$: The third derivative of $\sigma(u)$ is:
$$\sigma'''(u) = \frac{d}{du}(2S'(u) + uS''(u)) = 2S''(u) + (S''(u) + uS'''(u))$$
$$= 3S''(u) + uS'''(u)$$
This is a continuous function that decays to zero. Hence is bounded.

Therefore, $\sigma(u)$, $\sigma'(u)$, and $\sigma''(u)$ are all Lipschitz for Swish with $\beta = 1$.

**Non-vanishing Derivative:** The expected derivative $\mu_j$ is:
$$\mu_j = \mathbb{E}[\sigma'(u_j)] = \mathbb{E}[S(u_j) + u_j S'(u_j)]$$
$$= \mathbb{E}[S(u_j)] + \mathbb{E}[u_j S'(u_j)]$$

We evaluate each term:

For $\mathbb{E}[S(u_j)]$: The function $g(u) = S(u) - 1/2$ is an odd function. Since $u_j \sim N(0, \sigma_{u_j}^2)$ has a probability density function symmetric about 0, the expectation of any odd function of $u_j$ is 0. Thus, $\mathbb{E}[S(u_j) - 1/2] = 0$, which implies $\mathbb{E}[S(u_j)] = 1/2$.

For $\mathbb{E}[u_j S'(u_j)]$: The derivative of sigmoid, $S'(u) = S(u)(1 - S(u))$, is an even function: $S'(-u) = S(-u)(1 - S(-u)) = (1 - S(u))S(u) = S'(u)$. The product $h(u) = uS'(u)$ is an odd function, being the product of an odd function ($u$) and an even function ($S'(u)$). Since $u_j \sim N(0, \sigma_{u_j}^2)$ has a symmetric PDF about 0, $\mathbb{E}[u_j S'(u_j)] = 0$.

Combining these results:
$$\mu_j = 1/2 + 0 = 1/2$$

The value $1/2$ is a positive constant, independent of other parameters such as $d, n, m, \nu, \alpha$, or the specifics of $\Sigma$ (provided it is positive definite) and $w_j$ (provided $w_j \in \mathcal{S}^{d-1}$).

### B.1.6 Softplus Function

Let $\sigma(u) = \log(1 + e^u)$.

**Smoothness:** The Softplus function is $C^\infty$ for all $u \in \mathbb{R}$.
$$\sigma'(u) = \frac{e^u}{1 + e^u} = \text{sigmoid}(u)$$
$$\sigma''(u) = \frac{e^u}{(1 + e^u)^2} = \text{sigmoid}(u)(1 - \text{sigmoid}(u))$$

Both $\sigma'(u)$ and $\sigma''(u)$ exist for all $u \in \mathbb{R}$.

**Lipschitzness:** The lipschitzness follows from the boundedness and lipschitzness of sigmoid.

**Non-vanishing Derivative:** Following the argument presented for the Swish activation function, the mean is 0.5.

### B.2 Loss Function Derivatives

Let us see what this is for some common loss functions.

- For the Mean Squared Error (MSE) loss,

$$L(f(X)) = \frac{1}{2}\|f(X) - y\|^2 = \frac{1}{2}\sum_{i=1}^{n}(f(x_i) - y_i)^2 \quad \text{and} \quad L'(f(x)) = f(x) - y.$$

- For the Binary Cross Entropy (BCE) loss, we assume the network produces logits $z = f(X) \in \mathbb{R}^n$ with associated class-one probabilities $p = \text{sigmoid}(z) = \frac{1}{1+e^{-z}} \in \mathbb{R}^n$ computed component wise. Then, for given output data $y \in \{0, 1\}^n$,

$$L(f(X)) = -\sum_{i=1}^{n}\Big[y_i \ln(p_i) + (1 - y_i)\ln(1 - p_i)\Big], \quad L'(f(X)) = p - y = \text{sigmoid}(f(X)) - y.$$

- For the Hinge loss for binary classification with output data $y \in \{-1, 1\}^n$, $f(X) \in \mathbb{R}^n$ and

$$L(f(X)) = \sum_{i=1}^{n}\max\left(0,\, 1 - y_i\, f(x_i)\right).$$

Then $L'(f(X))$ is the vector whose $i$th entry is given by the subgradient

$$\frac{\partial L}{\partial f(x_i)} = \begin{cases} 0, & \text{if } y_i\, f(x_i) \geq 1, \\ -y_i, & \text{if } y_i\, f(x_i) < 1. \end{cases}$$

### B.3 Residue Concentration

1. Suppose the training labels satisfy $y_i = f_*(\mathbf{x}_i) + \xi_i$, where $f_*$ is Lipschitz and $\xi_i$ are i.i.d. subgaussian random variables. Then, for independent $W$ and $X$, lipschitz activation functions and for either the MSE or Binary Cross Entropy (BCE) loss the residues are subgaussian variables and satisfy this assumption.

2. For binary classification with the hinge loss, then since $a_i \sim \text{Unif}(\pm 1)$ we have with probability $1 - o(1)$ that at least a constant fraction of the data points satisfy $1 - y_i f(x_i) \geq 0$, and therefore $r_i = \pm 1$. As a result the assumption holds at initialization.

### B.4 $\beta$ Alignment

Here we consider `Sigmoid`, `ReLU`, `Tanh`, `ELU`, `Softplus`, and `Swish` activation functions. For each activation function, we consider three different loss functions - MSE, BCE, and Hinge. Then for for each activation and loss function combination, we consider $(\nu, \alpha) \in \{1/8, 3/8, 5/8\} \times \{0, 1/2\}$. This gives us 96 scenarios. We do each each scenario for the Mean Field and NTK scalings. For each scenario we let $\psi_1 = 0.75$ and $\psi_2 = 1.25$. We consider $n \in \{750, 1500, 2250, 3000, 3750\}$. We use triple index targets

$$f(x) = \text{sigmoid}(\beta_1^T x) + \tanh(\beta_2^T x) + \text{relu}(\beta_3^T x)$$

for three unit vectors $\beta_1, \beta_2, \beta_3$. For each value we do 50 trials to get the mean inner product $|\frac{1}{\sqrt{n}\|r\|}z^T r|$. Then we then estimate beta using linear regression.

Figure 9, presents the estimates $\beta$s. Here we see that $\beta$ has a mode around 1. Recall if $z_1, z_2$ are independent uniformly unit norm vectors. Then $z_1^T z_2 \sim d^{-1}$. Figure 9, however, that many $\beta$s are bigger than 1. This suggest $z, r$ are rapidly becoming orthogonal. Note that negative $\beta$s are cases, where the alignment improves, so $z, r$ are becoming parallel. Eventually, the inner product will saturate at 1 and $\beta$ should be close to zero. The reason we get negative $\beta$s is due to the limited range of $n$ used for the experiments.

## C Empirical Details

All code for the experiments can be found at Link.

The following details are common for all experiments.

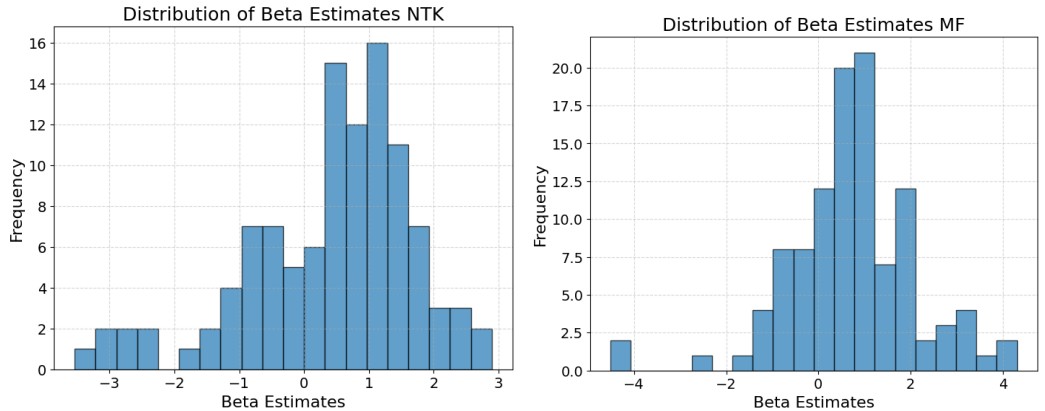

**Figure 9:** Estimated $\beta$ values

**Hardware:** All experiments were run on Google Colab using an A100.

**Data $X$:** We sampled $q$ uniformly randomly from the unit sphere and we used a diagonal $\hat{\Sigma}$.

**$\mu$ Estimation:** We estimate $\mu$ using 10000 samples.

**Targets:** The triple index model we used is as follows.

$$f(x) = \text{sigmod}(\beta_1^T x) + \tanh(\beta_2^T x) + \text{relu}(\beta_3^T x)$$

For three unit vectors $\beta_1, \beta_2, \beta_3$.

When using MSE loss, we let

$$y = f(x) + \varepsilon$$

for standard gaussian noise $\varepsilon$.

When using BCE loss, we use

$$y = f(x).$$

Note that these $y$ are not necessarily in $[0, 1]$. However, the BCE loss is still well defined.

When using Hinge loss,

$$y = \text{sign}(f(x) - 0.5).$$

Note this dataset can be imbalanced.

**Alignment determination:** To plot the red and blue lines in Figures 1,2,3,7,8, we use the following procedure. We let $B = S_1 + S_{12} + S_2$ (+ $S_3$ for the gradient penalty). Then we compute its leading left singular vectors for $B$. We then check if with $q$ and $X_B^T r$. Thus, how we get the associated singular value and we plot the corresponding lines.

### C.1 Figure 1

For non-isotropic $W$, we generate $W_S$ by sampling the rows i.i.d. from the unit sphere. We then introduce anisotropy, by adding $n^{-1/4} \mathbf{1} q^T$ to $W_S$ and then renormalizing to unit norm. This results in the weight concentrating around $q$.

### C.2 Figure 3

For Figure **(b)**, we generate $W_S$ by sampling the rows i.i.d. from the unit sphere. We then introduce anisotropy, by adding $n^{1/2} \mathbf{1} q^T$ to $W_S$ and then renormalizing to unit norm. This results in the weight concentrating around $q$.

For Figure **(c)**, we generate $W_S$ by sampling the rows i.i.d. from the unit sphere. Then we project onto the ortho-complement of $q$ and renormalize the rows.

For Figure **(d)**, we generate $W_S$ by sampling the rows i.i.d. from the unit sphere. We then let $W = W_S X^T X$ and renormalize the rows. This results in a $W$ that is highly dependent on $X$.

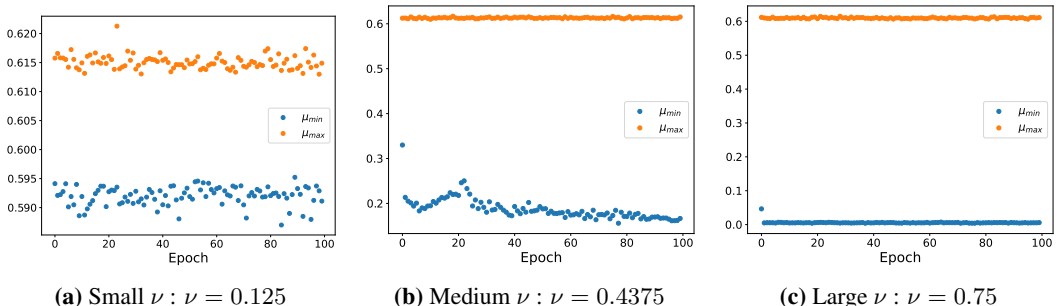

**(a)** Small $\nu$ : $\nu = 0.125$  **(b)** Medium $\nu$ : $\nu = 0.4375$  **(c)** Large $\nu$ : $\nu = 0.75$

**Figure 10:** Evolution of $\mu_{\min}$ and $\mu_{max}$ during training. Fixed parameters: MF scaling, Tanh activation, MSE loss, $\alpha = 0$.

### C.3 Figure 4

Here we use $\zeta = 0, \alpha = 0$. Hence applies for prior work from [3, 28].

We let $n \in \{100, 200, 300, 400, 500, 600, 700, 800\}$ and use $d = n/2$ and $m = n/3$.

### C.4 Later in Training Experiments

Here both network are initialized with the same weight matrix for both the inner and outer layers.

We use a step size of $\eta = \gamma_m^{-1}$. Additionally, after each iteration, we re-normalize the rows of $W$ to have unit norm.

For Figure 5**(c)**, the mean principal angle in the following quantity. Given orthonormal basis $u_1, \ldots, u_k$ and $v_1, \ldots, v_k$ for two subspaces, we form the matrix $A$ via

$$A_{ij} = u_i^T v_j$$

We the compute $\cos(\sigma_i(A))$. These are the principal angles between the subspaces. We then report the mean of angels.

### C.5 Real Data Experiments

**MNIST Dataset:** We load the standard MNIST dataset, divide by 256 to have all entries in $[0, 1]$. We use 1000 centered and flattened MNIST images to form $X \in \mathbb{R}^{1000 \times 784}$. We estimate $\nu \approx 0.784 > 1/2$. The data is highly ill-conditioned, suggesting a large effective $\alpha$.

**CIFAR Dataset:** We use $n = 1000$ CIFAR-10 training images, processed through a pretrained ResNet-18 (on ImageNet) to extract 512-dimensional penultimate-layer activations, forming $X \in \mathbb{R}^{1000 \times 512}$. We estimate $\nu \approx 0.3572 < 1/2$ and $\alpha \approx 0.6$.

Spcifically, the code for the transformations are as follows.

```
resnet18(weights=ResNet18_Weights.DEFAULT)

transform = transforms.Compose([
    transforms.Resize(224),
    transforms.ToTensor(),
    transforms.Normalize(mean=[0.485, 0.456, 0.406],  # ResNet defaults
                         std=[0.229, 0.224, 0.225])
])
```

## D  Later in Training

For Theorem 3.1 and Theorem 3.2 to apply beyond initialization, during training we require certain assumptions to hold. We begin by considering the common assumptions needed for both theorems.

1. Assumption 1 concerns the proportional regime and hence holds during training.

2. Assumption 2 concerns the data generation process and hence also holds during training.

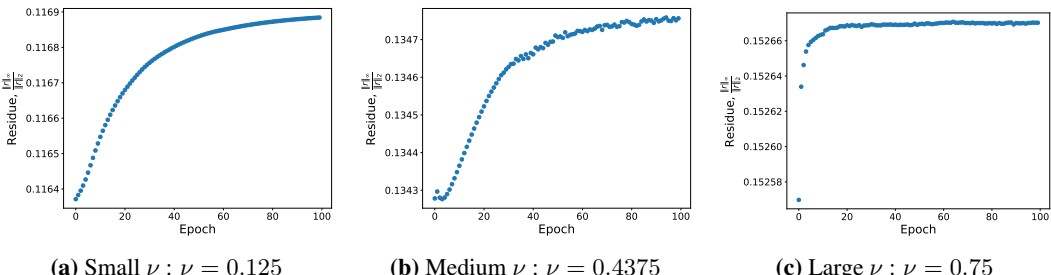

**(a)** Small $\nu$ : $\nu = 0.125$      **(b)** Medium $\nu$ : $\nu = 0.4375$      **(c)** Large $\nu$ : $\nu = 0.75$

**Figure 11:** Evolution of $\|r\|_\infty/\|r\|_2$ during training. Fixed parameters: MF scaling, Tanh activation, MSE loss, $\alpha = 0$.

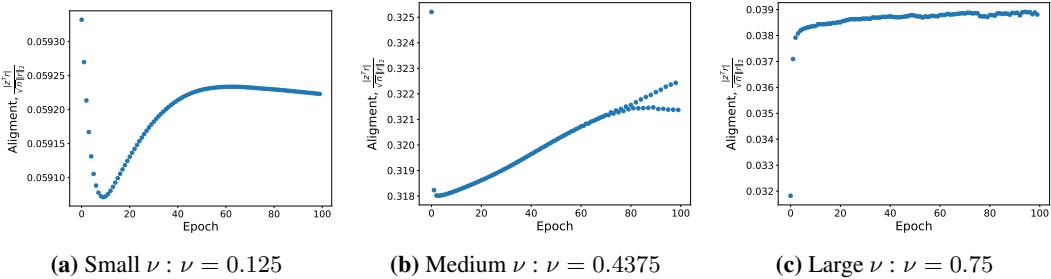

**(a)** Small $\nu$ : $\nu = 0.125$      **(b)** Medium $\nu$ : $\nu = 0.4375$      **(c)** Large $\nu$ : $\nu = 0.75$

**Figure 12:** Evolution of $|z^T r/(\sqrt{n}\|r\|_2)$ during training. Fixed parameters: MF scaling, Tanh activation, MSE loss, $\alpha = 0$.

3. Assumption 3 concerns the network initialization, and scaling, hence the assumptions on $a$ and $\gamma_m$ continue to hold during training. Moreover, through the use of weight normalization the assumption that the rows of $W$ are on the unit sphere also holds.

4. Assumption 4 concerns the activation function, namely its smoothness and Lipschitzness which of course also hold during training. However, it is not clear that the assumption on the non-vanishing gradient is satisfied. Despite this, we empirically verify as per Figure 10 that it does hold during training at least for small $\nu$. For moderate $\nu = 7/16$ we observe that $\mu_{min}$ appears to decrease, hence later in training this assumption may be violated. For large $\nu = 3/4$,the assumption only appears to hold for the first iteration. We remark that this results in the suppression of $S_1$ and $S_{12}$ but does not effect $S_2$ or $E$. As a result, we suspect that the data spike $q$ remains dominant.

5. Assumption 5. This is the assumption that

$$\frac{\|r\|_\infty}{\|r\|_2} = O\left(\frac{\log n}{\sqrt{n}}\right).$$

Figure 11 shows that while this ratio grows, the change is very small. Hence, we believe that this assumptions holds.

6. Assumption 6. This is about the alignment between $z$ and $r$. Figure 12 shows that while this ratio grows, the change is very small. Hence, we believe that this assumptions holds.

For the additional assumptions required for Theorem 3.1, clearly if the activation is $\mathcal{C}^2$ at initialization then it is also $\mathcal{C}^2$ throughout training. Finally, although clearly the independence of $W_t$ and $X$ is violated, due to the near constant gradient direction, (at least for the MF scaling) the correlation between $W$ and $X$ remains small.

