# OpenReview forum: "Low Rank Gradients and Where to Find Them"
_NeurIPS.cc/2025/Conference — NeurIPS 2025 poster_

### Official Review · Reviewer_k44k · 2025-06-21

**Clarity:** 3
**Significance:** 3
**Originality:** 3
**Rating:** 5
**Confidence:** 3

**Summary:**

This paper presents a theoretical and empirical investigation into the low-rank structure of gradients in two-layer neural networks without assuming data isotropy. Under the spike Gaussian ensemble with small, moderate, and large spike regimes, the high-dimensional asymptotic behaviors with proportional diverging n, m, d (as described in Assumption 1) are derived. It shows that gradients are dominated by two rank-one components related to the bulk data–residue and the input data structure. The influence of activation functions, scaling regimes (both mean field (MF) and neural tangent kernel (NTK) regimes), and regularization methods is also characterized.

**Questions:**

- How does the gradient influence the training process? Empirically, is it possible to simultaneously show the change in loss and the change in gradient rank during training? For example, Figure 6(b) shows a phase transition in gradient rank—does the loss exhibit any corresponding behavior?
- Could the authors provide a guideline or criterion for selecting different training strategies under varying conditions?

**Ethical Concerns:**

["NO or VERY MINOR ethics concerns only"]

**Final Justification:**

After reviewing the initial assessments, the other reviewers’ comments, and the authors’ rebuttal, I incline to recommend acceptance. The work extends prior analyses beyond isotropic settings, derives asymptotic results under a spiked Gaussian model, and characterizes the roles of activation, scaling regime (MF/NTK), and regularization. These extensions are novel relative to prior literature, and the theoretical findings are corroborated with simulations. In addition, the rebuttal was clear and addressed my concerns. Overall, the paper’s clarity, theoretical rigor, and extension of existing frameworks make it a valuable contribution to the theory of learning under non-isotropic data. I lean toward acceptance.

**Limitations:**

Yes

**Quality:**

3

**Strengths And Weaknesses:**

**Strengths:**
- The paper provides a rigorous theoretical framework for understanding low-rank gradients under anisotropic and ill-conditioned data.
- The paper analyzes how common regularizers (e.g. weight decay, noise, Jacobian penalty) affect gradient structure. Meanwhile, behaviors across both MF and NTK scalings are taken into account.

**Weaknesses:**
- The paper primarily focuses on gradients at or near initialization, with less on full training trajectories.
- Assumption 2 requires that the input data $x_i$ follow a (spike) multivariate Gaussian distribution.
- Only standard two-layer architectures are considered; extensions on the effects in CNNs, transformers, etc., are not explored.

---

> ### Author Rebuttal · Authors · 2025-07-29
>
> We thank the reviewer for their thoughtful and positive assessment.
>
> **Rebuttal TLDR:** While our theory is developed at (or very near) initialization for a 2 layer network, we explain below how the rank‑2 structure it reveals continues to shape learning dynamics and how the analysis can be broadened beyond the current Gaussian‑spike setting and to CNN setting.
>
> ### **Positives from the Review**
>
> We would also like to highlight the positives from the review
>
> * Develops a **rigorous asymptotic theory** that explains why gradients collapse to rank $\le 2$ under anisotropic (spiked) data.
> * **Unifies MF and NTK regimes** within the same analytical framework, showing how each scaling modulates the low‑rank structure.
> * **Dissects the impact of common regularizers** weight decay, isotropic noise, and Jacobian penalty, and links each to specific changes in gradient rank.
> * **Combines theory with targeted experiments**, validating the predicted phase transition in gradient rank across spike strengths.
>
> ### **Responses to the reviewers concerns**
>
> > The paper primarily focuses on gradients at or near initialization, with less on full training trajectories.
>
> We acknowledge this limitation. Establishing the rank‑2 structure at initialization is only a first step, but it is the prerequisite for any later‑time analysis; extending the theory along the entire trajectory is left for future work.
>
> > Assumption 2 requires that the input data $x_i$  follow a (spike) multivariate Gaussian distribution.
>
> The Gaussian assumption is mainly technical.
> * **Large‑spike regime:** our arguments rely only on sub‑Gaussian concentration and therefore carry over unchanged to generic sub‑Gaussian inputs, with a potential path toward sub‑exponential or even sub‑Weibull tails.
> * **Small‑spike regime:** the obstacle is the use of the Gaussian Poincaré inequality; since analogous inequalities hold for many log‑concave distributions, the same proof strategy should apply there as well.
>
> > Only standard two-layer architectures are considered; extensions on the effects in CNNs, transformers, etc., are not explored.
>
> We note that a convolutional network can be represented as a feed‑forward two‑layer network with a sparse, weight‑tied matrix $W$.  Assuming the convolutional filter $w$ is $\ell_2$‑normalized, our theory applies.
>
> Let $x\in\mathbb{R}^d$ be the input, $w=(w_1,\dots,w_k)\in\mathbb{S}^{k-1}$ the filter, and $a\in\mathbb{R}^m$ the second‑layer weights, where $m=d-k+1$.  Define
>
> $$
> W_{i,j}=
> \begin{cases}
> w_{\,j-i+1}, & 1\le j-i+1\le k,\\\\
> 0, & \text{otherwise},
> \end{cases}
> \qquad W\in\mathbb{R}^{m\times d},
> $$
>
> so the network output is
>
> $$
> y = a^{T}\sigma(Wx).
> $$
>
> Treating the entries of $W$ as independent parameters yields the unfolded gradient
>
> $$
> G = \frac{\partial L}{\partial W}\in\mathbb{R}^{m\times d},
> $$
>
> to which Theorems 3.1–3.2 apply, implying $\operatorname{rank}(G)=2$.
>
> Because $W$ is tied through $w\mapsto W$, the true filter gradient is
>
> $$
> \frac{\partial L}{\partial w_\ell}
> = \sum_{i=1}^m G_{i,i+\ell-1},
> \quad
> \ell=1,\dots,k.
> $$
>
> Writing $G = u^{(1)}(v^{(1)})^{T} + u^{(2)}(v^{(2)})^{T}$ with $u^{(r)}\in\mathbb{R}^m$ and $v^{(r)}\in\mathbb{R}^d$, and letting $\tilde{v}^{(r)}\_i = v^{(r)}\_{d-i+1}$, we have
>
> $$
> \nabla_w L
> = u^{(1)} * \tilde v^{(1)} + u^{(2)} * \tilde v^{(2)},
> $$
>
> so $\nabla_w L$ lies in a subspace of dimension at most $2$.  Preliminary CNN experiments confirm this prediction; we will include the plots in the appendix.
>
>
> > How does the gradient influence the training process?
>
> Using the learning‑rate prescription $\eta=\gamma_m^{-1}$, we consistently observe lower loss in the MF regime than in the NTK regime.
>
> For the run shown in Fig. 6 (b), we logged the loss alongside the gradient alignment. However, the loss curve remains smooth, so no dramatic visual “kink’’ appears.  We will add the combined plot to the supplement.
>
> We also conducted a small experiment to test how different “spikes’’ affect performance.  With $\nu=3/8$ we trained three networks that share the same architecture but differ in the first‑layer setup:
>
> * **ReLU** (data‑aligned spike)
> * **Sigmoid** (residue‑aligned spike)
> * **Sigmoid + Jacobian penalty** (data‑aligned spike)
>
> Inner‑layer weights were trained in the NTK regime for 100 epochs while freezing the outer layer.  Results:
>
> | variant | train loss (inner) | test loss (inner) |
> |---------|-------------------|-------------------|
> | Sigmoid (no reg.) | **lowest** | **lowest** |
> | ReLU | medium | medium |
> | Sigmoid + Jac. pen. | highest | highest |
>
> After subsequently fitting the outer layer (ridge regression), the ordering **reversed**:
>
> | variant | final train loss | final test loss |
> |---------|-----------------|-----------------|
> | Sigmoid + Jac. pen. | **lowest** | **lowest** |
> | ReLU | medium | medium |
> | Sigmoid (no reg.) | highest | highest |
>
> This illustrates that a residue‑aligned spike can speed early optimization but degrade generalization once the readout layer adapts.
>
>
> > Could the authors provide a guideline or criterion for selecting different training strategies under varying conditions?
>
> A pragmatic rule‑of‑thumb emerging from our analysis is:
>
> * **Leading eigendirection helpful for the task** → use a non‑saturating activation such as ReLU and/or add a Jacobian penalty to keep the spike data‑aligned.
> * **Leading eigendirection harmful / carries spurious signal** → use a smooth activation (sigmoid, GELU) and/or inject isotropic noise to push the spike into the residue.
>
> These choices control whether training emphasizes or suppresses the dominant data mode, offering a knob to match the assumed structure of the task.

---

> > ### Comment · Reviewer_k44k · 2025-08-05
> >
> > The authors' rebuttal has resolved most of my concerns. I'd like to keep the current score.

---

> > > ### Author Response · Authors · 2025-08-06
> > >
> > > We thank the reviewer for their feedback and continued support. If the reviewer has any other questions, we are happy to answer them.

---

### Official Review · Reviewer_4UNe · 2025-07-02

**Clarity:** 3
**Significance:** 2
**Originality:** 3
**Rating:** 4
**Confidence:** 4

**Summary:**

This paper investigates the low-rank structure of training-loss gradients in two-layer neural networks. Under a spiked data model, the authors analyze both mean-field and Neural Tangent Kernel (NTK) scaling regimes and demonstrate that the input gradient can be closely approximated by a rank-two matrix.

**Questions:**

Given the independence assumption between $X$ and $W$ in Theorem 3.1, why focus exclusively on the first training update? Can the analysis be extended to subsequent training steps where this independence no longer holds?

**Ethical Concerns:**

["NO or VERY MINOR ethics concerns only"]

**Final Justification:**

They answered all my questions.

**Limitations:**

Yes

**Paper Formatting Concerns:**

The reviewer has no formatting concerns.

**Quality:**

2

**Strengths And Weaknesses:**

## Strengths
- The manuscript is well-organized and clearly written.
- The theoretical results are precise, rigorous, and elegantly presented.
- Comprehensive empirical evaluations substantiate the theoretical insights.

## Weaknesses
- Theorem 3.1 relies on the strong assumption that data $X$ and weights $W$ are independent, a condition that holds only at the very first training step. It is unclear how—or if—this analysis extends to later stages of training when $X$ and $W$ become correlated.

---

> ### Author Rebuttal · Authors · 2025-07-29
>
> We thank the reviewer for the comments and feedback. In particular for highlighting:
>
> * The manuscript is well-organized and clearly written.
> * The theoretical results are precise, rigorous, and elegantly presented.
> * Comprehensive empirical evaluations substantiate the theoretical insights.
>
> We now address the reviewers concerns
>
> >  Why Theorem 3.1 matters even at step 1
> * It allows anisotropy in **both** the data \(X\) *and* the initial weights \(W\), and tolerates ill‑conditioned bulks—features absent from prior rank‑one analyses.
>
> * It unifies **mean‑field and NTK** scalings, whereas earlier work treats only one regime.
>
> Understanding this richer first‑step geometry is essential: it sets the initial condition for subsequent feature evolution.
>
> >  Extending beyond the first update
>
> 1. **Large‑spike regime.** Theorem 3.2, which governs large spikes, assumes *no* independence between $X$ and $W$.  Its rank‑two conclusion therefore holds throughout training.  In particular, in Appendix D (p. 37), we empirically verify that the rest of the assumptions still hold later in training.
>
> 2. **Small / moderate spikes.** At iteration $t$ we can draw a fresh mini‑batch $\tilde X_t$ that is independent of the current weights $W_t$.  This restores the required hypothesis  of Theorem 3.1.  Appendix D (p. 37) shows that the remaining conditions (residue concentration, alignment, etc.) still hold empirically, so the theorem applies to $(\tilde X_t,W_t)$ at every step.
> Resampling is a standard device in stochastic gradient methods and matches practical minibatch training.
>
> **Summary**
>
> Independence is needed only for Theorem 3.1 and only to control *small* spikes.  Large spikes are already covered by Theorem 3.2 for the full trajectory, and small spikes can be re‑analysed at any step via minibatch resampling.  We will add a remark making this explicit in the final version.

---

> > ### Comment · Reviewer_4UNe · 2025-08-06
> >
> > Thank you for your response. I went over the statement and the proof for Theorem 3.2. It doesn't require the dependence between $W$ and $X$. However, the proof only provides an upper bound for $E$ instead of $E_L$, and there should be an extra part $S_1$ inside $E_L$. Can you elaborate on this?
> >
> > Besides, I noticed that the key point of the proof is that you require $W$'s columns to be on the unit sphere. On the other hand,  the paper considers the GD updates. How can you ensure that both conditions are satisfied?
> >
> > Furthermore, in your experiments, you consider the vision models. Especially, you extract the hidden states before the last layer and use those as the input $X$. But in the theory part, you consider a two-layer NNs with fixed weights $\gamma_m$ and learnable parameters $W$. Can you give more explanation on this?  Like, why are they related in some sense?
> >
> > A small typo is that you didn't add $\gamma_m$ in your gradient formula in Proposition 2.1. Besides, you didn't consider the effect of the regularization in your theory. It's better to get rid of the term in your gradient formula.

---

> ### Author Response · Authors · 2025-08-07
>
> We thank the reviewer for their comment. We answer the reviewers questions below.
>
> > However, the proof only provides an upper bound for $E$ instead of $E_L$, and there should be an extra part $S_1$ inside $E_L$. Can you elaborate on this?
>
> In the proof for Theorem 3.2 we focused on the terms whose scaling is different between Theorem 3.1 and Theorem 3.2
>
> In particular, Lemma A.1 still tells us that $\|\|S_1\|\|_2 \lesssim \sqrt{m} \gamma_m \|\|r\|\|_2 n^{-1/2}$. The proof for Theorem 3.2 gives us that $\|\|E\|\|_2 \lesssim \sqrt{m} \gamma_m \|\|r\|\|_2$. Hence here the $S_1$ is much smaller.
>
> This then gives us the bound that
> $$\|\|E_L\|\|_2 = \|\|E + S_1\|\|_2 = O(\sqrt{m} \gamma_m \|\|r\|\|_2)$$
>
> As discussed in 191-195, if certain conditions hold, then we are guarateed to have the $S_2$ spike escape the bulk and hence the gradient will be rank 1 **not** rank 2. This is empirically verified in Figure 3.
>
> > Besides, I noticed that the key point of the proof is that you require $W$‘s columns to be on the unit sphere. On the other hand, the paper considers the GD updates. How can you ensure that both conditions are satisfied?
>
> The current Theorem 3.2 is presented as such for simplicity. However, this is not needed for Theorem 3.2 ($\nu > 1/2$).
>
> For the Theorem 3.1 ($\nu < 1/2$), we need the assumption to get explicit tighter bounds on $E$ and $S_2$, as these terms depend on $\sigma’_{\perp}(XW^T)$. In particular, we need the entries of $XW^T$ to be gaussian with equal variance when conditioned on $W$. Hence we need all rows of $W$ to be unit norm.
>
> However, for Theorem 3.2 ($\nu > 1/2$), we do not use these concentration inequalities. Instead, we note that since $\sigma$ is Lipschitz, $\sigma’$ is bounded, hence $\sigma’_{\perp}(XW^T)$ has uniformly bounded entries. Thus, the operator norm cannot be larger than $O(n)$. We use this in 652 and 660 in the proof of Theorem 3.2.
>
> Furthermore, when we run experiments, this assumption is still satisfied for later steps. As discussed in line 199, we employ the common weight normalization technique from [27]. This projects the weights back onto the sphere.
>
> [27] Tim Salimans and Durk P Kingma. Weight normalization: A simple reparameterization to
> accelerate training of deep neural networks. In Advances in Neural Information Processing Systems
>
> > Furthermore, in your experiments, you consider the vision models. Especially, you extract the hidden states before the last layer and use those as the input $X$. But in the theory part, you consider a two-layer NNs with fixed weights $\gamma_m$ and learnable parameters $X$. Can you give more explanation on this? Like, why are they related in some sense?
>
> These experiments were to highlight that our theory provides intuition even in cases where the theory does not strictly apply.
> In particular, let $F$ be a pretrained network and let $F_\ell$ be its output at some layer $\ell$. Then given image data $Z$, we consider $X = F_\ell(Z)$.
>
>
> We then use this $X$ as the input to train another network $f(X) = a^T \sigma(WX^T)$ and show that the gradient of $W$ is as predicted and that regularization has the predicted qualitative effect as well.
>
>
> > A small typo is that you didn’t add $\gamma_m$ in your gradient formula in Proposition 2.1. Besides, you didn’t consider the effect of the regularization in your theory. It’s better to get rid of the term in your gradient formula.
>
> We thank the reviewer for pointing out this typo. We shall fix this.
>
> We have theory results on regularization. While Theorem 3.1 and Theorem 3.2 do not talk about the regularization, Propositions 4.1, 4.2, 4.3 analyze the effect of regularization. In particular, Propositions 4.1 and 4.3 consider regularizers of the form in Proposition 2.1. Hence we presented the gradient jointly.
>
>
> We hope that our answers have cleared any doubts the reviewer had. If the reviewers has further questions. We would be happy to answer them.

---

### Official Review · Reviewer_y7gX · 2025-07-03

**Clarity:** 3
**Significance:** 2
**Originality:** 2
**Rating:** 4
**Confidence:** 4

**Summary:**

This paper studies the implicit gradient structure in two-layer feedforward neural networks under generalized data assumptions (anisotropy) and independence assumptions on weight initializations. When the input data distribution follows a rank-one Gaussian *spiked* model with a (possibly anisotropic) bulk covariance, then under some technical conditions, the loss gradient will exhibit a specific low-rank structure in weight space. In particular, it is shown that a rank-two spike emerges, composed of a "data'' spike which is aligned to the spike in the input covariance, and a "residue'' spike which is aligned to the loss derivative (e.g. $y - f(x)$ in the case of square loss). It is then shown the relative strength of the two spikes is determined essentially by three things: 1. the input spike strength, 2. the regularity properties of the input activation, and 3. the "scale'' parameter, e.g. Neural Tangent Kernel or Mean-Field regimes. These predictions are corroborated in simulation. Lastly, the effect of various regularizers is studied.

**Questions:**

Beyond responding or clarifying the points in Weaknesses above, I have the following clarification questions:

Can the bounds be applied inductively for any-iteration guarantee? I do not see clearly stated whether the intention is for the theoretical bounds to hold beyond early-training (for example, Assumption 3 implies the output layer weights are random and not updated).

What are predictions for how the observed structure behaves beyond a precisely rank-one spike, such as for low-rank or approximate spikes with rapidly decaying (but not bulk-separated) singular values? For example, for $r = O(1)$ spikes of sufficiently large and comparable strength, how many spikes do we expect to see in the gradient?

Why is the spike strength $\zeta$ defined as a function of the data/batch size $n$? Maybe I'm missing something, but it would seem more natural to be a function of the data dimension $d$, as it is a population-level quantity.

**Ethical Concerns:**

["NO or VERY MINOR ethics concerns only"]

**Final Justification:**

This paper presents some deep learning theory demonstrating the emergence of a low-rank structure in the gradient of two-layer networks in data-anisotropic settings. The paper presents some new results that may be interesting to the field, and therefore I lean toward acceptance (4). My main concern with the paper is that it does not concretely tie this new phenomena to generalization behavior, and thus does not yet present actionable takeaways.

**Limitations:**

Yes

**Quality:**

3

**Strengths And Weaknesses:**

Overall, I think the paper is clearly presented and concretely establishes some novel phenomena in the learning theory of neural networks. Therefore, I lean toward acceptance of the paper. My main concerns with the paper lie with the breadth and relevance of the results in the paper.

## Strengths

Compared to prior work studying feature learning on two-layer networks via gradient descent, this paper generalizes the setting in various ways. Notably, key prior work in feature learning considers isotropic gaussian inputs, and typically considers 1-step gradient updates from i.i.d. weight initializations. This work extends to a rank-one spiked model with anisotropic bulk under a general loss function, and computing the gradient at a weight matrix with possibly dependent entries. The key insight is that this added complexity leads to a quantifiable phenomenon in the residue and data spikes in the gradient, which otherwise does not manifest in the more ideal settings in prior work. I think documenting this phenomenon and providing some tools for analyzing feature evolution in broader settings are valuable.

The other technical assumptions are justified either from empirical observation or theoretical derivations in the appendix, which is appreciated.

## Weaknesses

In general, my main concern with the paper, tying back to prior work on feature learning, is that it is not clear to me what demonstrating the rank-two structure (and how it interacts with choice of activation function and regularization) implies about *feature learning* itself, or similarly learning performance. For example, it is not clear to me after reading the paper whether both spikes are useful for learning, or if the data spike can be spurious, such as in the prototypical single-index regression setting.

On that note, I am not sure if the raw gradient (and ipso facto gradient descent) is necessarily a good object to study for feature learning in anisotropic settings. For example, [1] and [2] demonstrate that SGD can be severely suboptimal when the data distribution is anisotropic, proposing forms of preconditioning to alleviate the poor convergence of SGD. From my reading [2] considers a two-layer network setting. Now, given a generic anisotropic gaussian input and a single-index target, the proposed algorithm is to precondition the weight gradient by the *inverse (sample $\approx$ population) covariance*. Therefore, it seems therein the authors recover an approximately *rank-one* gradient as in the seminal paper [3] (which considers isotropic inputs). If one places the spiked anisotropic covariance into the set-up of [2], it seems that a dichotomy occurs: the raw gradient should exhibit a rank-two structure as proven here but aligns very poorly with the correct direction, whereas a provably adjusted gradient recovers the *rank-one* structure from prior single-index learning papers.

To re-iterate: it seems in the general anisotropic setting, the raw gradient can be very ill-aligned to a descent direction, and existing alternate algorithms that improve on SGD's performance need not respect the rank-two spike structure. This is even more so when considering popular adaptive descent methods that alter the gradient in a non-linear fashion such as Adam and Muon, hence my concerns about the ultimate relevance of these observations.

[1] Amari et al, "When does preconditioning help or hurt generalization?"

[2] Zhang et al. "On The Concurrence of Layer-wise Preconditioning Methods and Provable Feature Learning"

[3] Ba et al. "High-dimensional asymptotics of feature learning: How one gradient step improves the representation"

---

> ### Author Rebuttal · Authors · 2025-07-29
>
> Thank you very much for your thoughtful and constructive review.  We appreciate your positive assessment of the paper’s contributions and your insightful questions, particularly regarding the relevance of the observed rank-two gradient structure for feature learning.
>
> ### Positives
>
> * **Generalization beyond prior work:** The paper extends existing feature learning analyses from isotropic Gaussian inputs and independent weights to a rank-one spiked covariance model with anisotropic bulk, under general loss functions and non-i.i.d. weight initializations.
>
> * **Novel gradient structure:** It identifies a new rank-two gradient phenomenon, with distinct residue- and data-aligned spikes, offering a richer understanding of feature evolution that does not arise in prior simplified settings.
>
> * **Sound theoretical and empirical grounding:** The technical assumptions are well-justified (empirically and analytically), and simulations corroborate the theoretical predictions.
>
> ### Concerns of the reviewer.
>
> > On that note, I am not sure if the raw gradient (and ipso facto gradient descent) is necessarily a good object to study for feature learning in anisotropic settings.
>
> We thank the reviewer for pointing us to papers [1] and [2], which indeed raise important considerations about optimization dynamics in anisotropic settings.
>
> However, we respectfully disagree with the assertion that studying the raw gradient is a “good object to study for feature learning in anisotropic settings”. On the contrary, we believe that both [1] and [2] reinforce the value of understanding the structure of unpreconditioned gradients:
>
> **Preconditioning need not kill rank 2.**
>
> In [2] the feature‑learning update involves $P_g$ and $Q_G$, both positive‑semidefinite.  A preconditioner that acts similarly on the residue‑ and data‑directions will *retain* the rank‑two structure.  Hence knowing the raw gradient’s geometry is still informative when designing or analyzing preconditioned algorithms.
>
> **Need Dual Understanding to Isolate Important Algorithmic Aspects**
>
> Second, while it is true that preconditioned SGD and adaptive methods like Adam may accelerate convergence or improve alignment in some settings, a theoretical understanding of why and when this occurs necessarily begins with understanding the raw gradient. Our work aims to provide this foundational understanding.
>
> **Beyond Single Index Model**
>
> The alignment metric in [2] is tailored to single‑index models.  Our labels $y=f(x)$ are only assumed Lipschitz, so they may depend on *many* directions.  In this broader setting it is unclear whether collapsing to rank 1 is optimal; the second spike can carry additional tasks‑relevant signals that single‑index theory neglects.
>
> **Broader scope than [2].**
>
> The core theorems of [2] (3.6 & 3.9) are themselves *gradient‑structure* results.  Theorem 3.9 is a **special case** of our Theorem 3.1: it is limited to mean‑field (MF) scaling, MSE loss, isotropic first‑layer weights, and a well‑conditioned covariance.  Our analysis covers both NTK *and* MF regimes, arbitrary losses, data‑dependent or anisotropic weights, and ill‑conditioned bulks via the spectral‑decay exponent $\alpha$.  Consequently we capture phenomena, e.g. the emergence of a **data‑aligned spike**, that [2] cannot see.
>
> **Rank‑two can beat rank‑one even in single‑index cases.**
>
> A mis‑aligned rank‑one update is not necessarily better than a rank‑two update whose second direction compensates for the misalignment.  Our framework provides the tools to analyse this possibility, which [2] leaves unexplored.
>
> **Regularization**
>
> Additionally, if the gradient is misaligned, other ways of fixing it besides natural gradients and Adam could be the use of regularization. Hence our study about the effect of regularization on the gradient is also important.
>
> **Comparison with [1]**
>
> Finally, we briefly comment on [1]. While that paper focuses on linear models and assumes well-conditioned data, its results are more nuanced than they may first appear. For example, Figure 3 of [1] shows that gradient descent (GD) initially reduces error faster than natural gradient descent (NGD), and that early-stopped GD can outperform early-stopped NGD. This suggests that the initial descent direction of GD, precisely the object we study, can be quite effective, and possibly superior in practice. Thus, understanding the structure of GD's updates remains valuable, even in light of competing optimization schemes.
>
> **Summary**
>
> In summary, this paper aims to provide a systematic characterization of the raw gradient in anisotropic, spiked settings. We view this as a necessary step toward understanding and potentially improving feature learning in practical neural networks, whether by standard SGD or by more sophisticated algorithms.
>
> > In general, my main concern with the paper, tying back to prior work on feature learning, is that it is not clear to me what demonstrating the rank-two structure (and how it interacts with choice of activation function and regularization) implies about feature learning itself, or similarly learning performance. For example, it is not clear to me after reading the paper whether both spikes are useful for learning, or if the data spike can be spurious, such as in the prototypical single-index regression setting.
>
> This is a great question that is currently being explored by the authors.
>
> Building on Moniri et al. (2024), we show that the learned feature matrix satisfies
>
> $$  F\_1 = F\_0 + \sum\_{k=0}^L \begin{bmatrix} c\_k^{\tau\_1} \\\\ \vdots \\\\ c\_k^{\tau\_m} \end{bmatrix} \mathbf{1}\_n^T \circ \left[ (1-\lambda \eta) \tilde{X}\_S W\_0^T + \eta \tilde{X}(S\_1 + S\_{12} + S\_2)\right]^{\circ k} + \Delta $$
>
> where $\Delta$ is small.  Whereas Moniri et al.’s Theorem 3.1 shows that features become **univariate polynomials of the spike, independent of weights**, our formula reveals **multivariate polynomials of both spikes and the evolving weight matrix** in the anisotropic setting.  This richer dependence explains why regularisers that favour a data‑aligned spike can improve generalisation after the outer layer is trained.
>
> We also conducted a small experiment to test how different “spikes’’ affect performance.  With $\nu=3/8$ we trained three networks that share the same architecture but differ in the first‑layer setup:
>
> * **ReLU** (data‑aligned spike)
> * **Sigmoid** (residue‑aligned spike)
> * **Sigmoid + Jacobian penalty** (data‑aligned spike)
>
> Inner‑layer weights were trained in the NTK regime for 100 epochs while freezing the outer layer.  Results:
>
> | variant | train loss (inner) | test loss (inner) |
> |---------|-------------------|-------------------|
> | Sigmoid (no reg.) | **lowest** | **lowest** |
> | ReLU | medium | medium |
> | Sigmoid + Jac. pen. | highest | highest |
>
> After subsequently fitting the outer layer (ridge regression), the ordering **reversed**:
>
> | variant | final train loss | final test loss |
> |---------|-----------------|-----------------|
> | Sigmoid + Jac. pen. | **lowest** | **lowest** |
> | ReLU | medium | medium |
> | Sigmoid (no reg.) | highest | highest |
>
> This illustrates that a residue‑aligned spike can speed early optimization but degrade generalization once the readout layer adapts.
>
>
> > Can the bounds be applied inductively for any-iteration guarantee? I do not see clearly stated whether the intention is for the theoretical bounds to hold beyond early-training (for example, Assumption 3 implies the output layer weights are random and not updated).
>
> Yes in the following way.
>
> *Large‑spike regime:* Theorem 3.2 does **not** depend on weight–data independence, so its conclusions hold for *all* iterations as long as the other assumptions continue to hold, which we verify empirically in App. D (p. 37).
>
> *Small / moderate spikes:* At step $t$ we can draw a fresh mini‑batch $\tilde X_t$ that is independent of the updated weights $W_t$.  Independence is then restored and the remaining assumptions still hold (again confirmed in App. D).  Hence the rank‑two guarantee can be applied inductively throughout training.
>
>
> > What are predictions for how the observed structure behaves beyond a precisely rank-one spike, such as for low-rank or approximate spikes with rapidly decaying (but not bulk-separated) singular values?
>
> If the population covariance has $k$ well‑separated spikes, our theory predicts $k+1$ spikes in the gradient matrix.  Their relative magnitudes follow exactly the same formulas as in Theorems 3.1 (small/mid spikes) and 3.2 (large spike), evaluated at the corresponding eigenvalues.
>
>
> > Why is the spike strength defined as a function of the data/batch size? Maybe I'm missing something, but it would seem more natural to be a function of the data dimension, as it is a population-level quantity.
>
> Since we consider the proportional scaling limit $n^\nu = \Theta(d^\nu)$ they are equivalent.

---

> > ### Comment · Reviewer_y7gX · 2025-08-05
> >
> > I thank the authors for their clarifications --- I believe a different reviewer may have accidentally spoken for me earlier. I maintain my overall positive evaluation and lean acceptance.
> >
> > An overall concern remains, which is that the paper as it currently stands doesn't express cleanly how the emergent rank-two structure tangibly affects final generalization, which limits its actionable takeaways. Also, to make an earlier point clear, I meant that studying gradient descent need not be the dogmatic algorithm of choice. Certainly, by nature of our current theoretical tools, we typically start with the gradient object, but the point of mentioning those papers was to elucidate that instead of trying to understand how **GD** can perform, one may instead choose a different descent method; **linear preconditioning** such as natural gradient / KFAC etc. may be nice to study if just for analytical convenience. Any fine-grained insights about the gradient / GD become more tenuous when using non-linear operations that are popular in practice such as Adam or more recently Muon, which remains a largely open issue in the area.

---

> > > ### Author Response · Authors · 2025-08-06
> > >
> > > Thank you for your continued engagement and positive evaluation.
> > >
> > > We agree that gradient descent need not be the algorithm of choice with other descent methods available. Our goal is not to advocate for raw GD per se, but to use the structure of the gradient update as a lens into how data anisotropy, initialization, and regularization shape feature evolution, especially in the early phases of training. While generalization under adaptive methods like Adam or Muon remains an open challenge, we view our results as a baseline: the emergence of a rank-two structure offers a concrete phenomenon that future algorithms may choose to preserve, exploit, or correct. We hope this contributes to building more principled foundations for the analysis of other practical methods.
> > >
> > > We thank you again for your thoughtful feedback.

---

> ### Comment · Reviewer_Y78K · 2025-08-03
>
> The rebuttal has solved most of the questions. I still keep my score to weakly accept this paper.

---

### Official Review · Reviewer_Y78K · 2025-07-04

**Clarity:** 3
**Significance:** 3
**Originality:** 4
**Rating:** 4
**Confidence:** 5

**Summary:**

This paper investigates the low-rank structure in gradients of the hidden-layer weights in two-layer neural networks. It considers a spiked covariance model with anisotropic and ill-conditioned bulk, extending the results of existing literature. The authors demonstrate that the gradient matrix is often approximately rank two. The paper characterizes how their relative influence is modulated by the data spectrum (via parameters ν and α), network scaling regimes, activation function, and regularization techniques (e.g., weight decay, input noise, Jacobian penalties). Theoretical predictions are validated through a comprehensive suite of experiments on both synthetic data and real-world embeddings (MNIST, CIFAR-10).

**Questions:**

See the Weakness part.

**Ethical Concerns:**

["NO or VERY MINOR ethics concerns only"]

**Final Justification:**

I have checked the author's rebuttal and other reviewers' comments, and I still tend to weakly accept this paper.

**Limitations:**

yes

**Quality:**

3

**Strengths And Weaknesses:**

Strengths

- The paper consider the gradient structure of two-layer NN by relaxing the usual isotropy assumptions on
the training data and parameters. The identification and characterization of a dominant rank-two gradient structure under general anisotropic data conditions appears to be novel. It sheds light on why and how low-rank structures persist under realistic data conditions, which is critical for interpreting deep learning models.
- The theoretical analysis is mathematically rigorous, particularly Theorems 3.1 and 3.2, which give precise scaling laws for the emergence of low-rank gradient structure, though I did not check all the proofs completely. The paper covers both mean-field and NTK scaling, expanding the applicability of results. Regularization effects are treated with nuance and technical depth (e.g., Propositions 4.1–4.4). Experiments are well-designed and carefully interpreted, providing convincing empirical support.

Weaknesses

- Limited by One-Step Gradient Analysis. Most of the theoretical results are framed around single-step or initialization-time analysis. While longer-term training dynamics are partially addressed empirically (e.g., Section 3.4), the analytical extension beyond early training is missing.

- Real-World Implications. 1) Although experiments on CIFAR and MNIST are included, these use fixed embeddings and do not reflect end-to-end training. 2) The paper will be improved if additional discussion could be added regarding practical implications (e.g., how this informs data preprocessing, architecture design or optimizer design).

---

> ### Author Rebuttal · Authors · 2025-07-29
>
> Thank you for your thoughtful and constructive review.  We are particularly grateful for your recognition of the paper’s strengths:
>
> ### **Positives**
>
> * **Novelty and generality:** you highlight the originality of our rank‑two gradient analysis under anisotropic, ill‑conditioned data, an essential generalisation of prior work.
>
> * **Theoretical depth:** you note the mathematical rigour of Theorems 3.1, 3.2 and the nuanced regularisation analysis.
>
> * **Experimental care:** you find the experiments well‑designed and consistent with theory on both synthetic data and real‑world embeddings.
>
> Below we address your main concerns.
>
> ### **Reviewer Concerns**
>
> > One step Gradient
>
> We agree that extending theoretical insights beyond the first gradient step is a crucial direction. While our main theorems are stated at initialization, the following mechanisms allow our analysis to remain relevant later in training:
>
> **Large‑spike regime (Thm. 3.2).** Because Thm. 3.2 does *not* rely on independence between data and weights, its conclusions hold after several training steps.  Appendix D (p. 37) verifies that the remaining assumptions continue to hold during training, so the rank‑two structure persists.
>
> **Resampling argument for Theorem 3.1:** In the small-to-moderate spike setting, Theorem 3.1 does require $W$ and $X$ to be independent. However, this requirement can be satisfied at iteration $t$ by resampling a new batch $\tilde{X}_t$, which is independent of the current weights $W_t$. In Appendix D (page 37), we empirically verify that the remaining assumptions (e.g., residue concentration and alignment) continue to hold later in training. This suggests that Theorem 3.1 still applies.
>
> > Practical Takeaway
>
> Note our current experiments on CIFAR-10 use fixed embeddings. However, we use the MNIST dataset with minimal whitening.
>
> We appreciate the request to sharpen the practical relevance of our findings. Although a full end‑to‑end treatment is future work, the present analysis already yields concrete insights.
>
> A pragmatic rule‑of‑thumb emerging from our analysis is:
>
> * **Leading eigendirection helpful for the task** → use a non‑saturating activation such as ReLU and/or add a Jacobian penalty to keep the spike data‑aligned.
> * **Leading eigendirection harmful / carries spurious signal** → use a smooth activation (sigmoid, GELU) and/or inject isotropic noise to push the spike into the residue.
>
> These choices control whether training emphasizes or suppresses the dominant data mode, offering a knob to match the assumed structure of the task.
>
> **Extension of feature‑learning dynamics.** Building on Moniri et al. (2024), we show that the learned feature matrix satisfies
>
> $$  F\_1 = F\_0 + \sum\_{k=0}^L \begin{bmatrix} c\_k^{\tau_1} \\\\ \vdots \\\\ c\_k^{\tau_m} \end{bmatrix} \mathbf{1}\_n^T \circ \left[ (1-\lambda \eta) \tilde{X}\_S W\_0^T + \eta \tilde{X}(S\_1 + S\_{12} + S\_2)\right]^{\circ k} + \Delta $$
>
> where $\Delta$ is small.  Whereas Moniri et al.’s Theorem 3.1 shows that features become **univariate polynomials of the spike, independent of weights**, our formula reveals **multivariate polynomials of both spikes and the evolving weight matrix** in the anisotropic setting.  This richer dependence explains why regularisers that favour a data‑aligned spike can improve generalisation after the outer layer is trained.
>
> We also conducted a small experiment to test how different “spikes’’ affect performance.  With $\nu=3/8$ we trained three networks that share the same architecture but differ in the first‑layer setup:
>
> * **ReLU** (data‑aligned spike)
> * **Sigmoid** (residue‑aligned spike)
> * **Sigmoid + Jacobian penalty** (data‑aligned spike)
>
> Inner‑layer weights were trained in the NTK regime for 100 epochs while freezing the outer layer.  Results:
>
> | variant | train loss (inner) | test loss (inner) |
> |---------|-------------------|-------------------|
> | Sigmoid (no reg.) | **lowest** | **lowest** |
> | ReLU | medium | medium |
> | Sigmoid + Jac. pen. | highest | highest |
>
> After subsequently fitting the outer layer (ridge regression), the ordering **reversed**:
>
> | variant | final train loss | final test loss |
> |---------|-----------------|-----------------|
> | Sigmoid + Jac. pen. | **lowest** | **lowest** |
> | ReLU | medium | medium |
> | Sigmoid (no reg.) | highest | highest |
>
> This illustrates that a residue‑aligned spike can speed early optimization but degrade generalization once the readout layer adapts.

---

> > ### Comment · Reviewer_Y78K · 2025-08-03
> >
> > The authors' rebuttal has solved most of my concerns. I still keep the score to weakly accept this paper.

---

> > > ### Author Response · Authors · 2025-08-04
> > >
> > > We thank the reviewer for their feedback. If the reviewer has any other questions we are happy to answer them.

---

### Decision · Program_Chairs · 2025-09-17

**Decision:**

Accept (poster)

**Comment:**

The paper considers two-layer neural networks under both mean-field and NTK scaling regimes and shows that the gradient of the training loss exhibits an approximately low-rank structure. This analysis relaxes the usual assumptions of isotropic data and parameters, as well as their independence. In particular, when the data follow a multivariate normal distribution with a rank-one spiked anisotropic covariance and an ill-conditioned bulk, the gradient is theoretically shown to be approximately rank-two, with its two dominant components aligned with the bulk residue and the input data spike. The relative influence of these components is modulated by the data spectrum, network scaling regime, activation function, and regularization techniques. The theoretical claims are supported by experiments.

The reviewers agree that the paper makes a meaningful theoretical contribution, noting that:
1) the insights into early training dynamics are interesting, while generalizing the commonly considered conditions for the data, weights, and their dependence, and considering both MF and NTK regimes, as well as the implications of regularization techniques
2) the theoretical results are rigorous, sound, well-presented, and well-supported by the experiments
3) the paper is well-written with potential for further impact.

A few concerns were raised, particularly regarding the implications of the analysis on feature learning and generalization, the fact that the analysis focuses on early training, and some technical clarifications. Most of these points were addressed in the rebuttal. I therefore recommend acceptance and encourage the authors to consider the feedback and update the final version of the paper accordingly.